# Evaluating the Robustness of Time Series Anomaly and Intrusion Detection Methods against Adversarial Attacks

## Abstract

Time series anomaly and intrusion detection are extensively studied in statistics, economics, and computer science. Over the years, numerous methods have been proposed for time series anomaly and intrusion detection using deep learning-based methods. Many of these methods demonstrate state-of-the-art performance on benchmark datasets, giving the false impression that these systems are robust and deployable in practical and industrial scenarios. In this paper, we demonstrate that state-of-the-art anomaly and intrusion detection methods can be easily fooled by adding adversarial perturbations to the sensor data. We use different scoring metrics such as prediction errors, anomaly, and classification scores over several public and private datasets belong to aerospace applications, automobiles, server machines, and cyber-physical systems. We evaluate state-of-the-art deep neural networks (DNNs) and graph neural networks (GNNs) methods, which claim to be robust against anomalies and intrusions, and find their performance can drop to as low as 0% under adversarial attacks from Fast Gradient Sign Method (FGSM) and Projected Gradient Descent (PGD) methods. To the best of our knowledge, we are the first to demonstrate the vulnerabilities of anomaly and intrusion detection systems against adversarial attacks. Our code is available here: `https://anonymous.4open.science/r/ICLR298`

## 1 Introduction

Machine learning and deep learning have profoundly impacted numerous fields of research and society over the last decade (LeCun et al., 2015; Goodfellow et al., 2016). Medical imaging (Litjens et al., 2017), speech recognition (Kumar et al., 2018), and smart manufacturing systems (Wang et al., 2018) are a few of these areas. With the proliferation of smart sensors, massive advances in data collection and storage, and the ease with which data analytics and predictive modeling can be applied, multivariate time series data obtained from collections of sensors can be analyzed to identify regular patterns that can be interpreted and exploited. Numerous researchers have been interested in time series anomaly and intrusion detection (Pang et al., 2021; Khraisat et al., 2019). For instance, time series anomaly detection methods are used in the aerospace industry for satellite health monitoring, while intrusion detection methods are employed in the automobile industry for in-vehicle controller area networks. These deep neural network-based solutions outperform the competition on a variety of benchmark datasets. However, as deep learning became more prevalent, researchers began to investigate the vulnerability of deep networks, particularly to adversarial attacks. In the context of image recognition, an adversarial attack entails modifying an original image in such a way that the modifications are nearly imperceptible to the human eye (Yuan et al., 2019). The modified image is referred to as an adversarial image, as it will be classified incorrectly by the neural network, whereas the original image will be classified correctly. One of the most well-known real-world attacks involves manipulating the image of a traffic sign in such a way that it is misinterpreted by an autonomous vehicle (Eykholt et al., 2018). The most common type of attack is gradient-based, in which the attacker modifies the image in the direction of the gradient of the loss function relative to the input image, thereby increasing the rate of misclassification (Yuan et al., 2019; Goodfellow et al., 2014; Madry et al., 2017).

While adversarial attacks have been extensively studied in the context of image recognition, they have not been extensively investigated for anomaly and intrusion detection systems. It is surprising given the increasing popularity of deep learning models for classifying time series (Ma et al., 2018; Zheng et al., 2017; Wang et al., 2017).

Additionally, adversarial attacks are a possibility in a large number of applications that require the use of time series data. For instance, Figure 1 (top) depicts the original and perturbed time series for the Korean Aerospace Research Institute's KOMPSAT-5 satellite (KARI). The prediction error (see Figure 1, bottom) is generated by the "Convolutional LSTM with Mixtures of Probabilistic Principal Component Analyzers" (CLMPPCA) method, which is currently deployed at KARI, to predict anomalies. While CLMPPCA accurately predicts the anomaly for the original time series, adding small perturbations in the form of FGSM and PGD attacks causes the entire input samples to be classified as an anomaly. This attack can have a severe impact on the satellite health monitoring system.

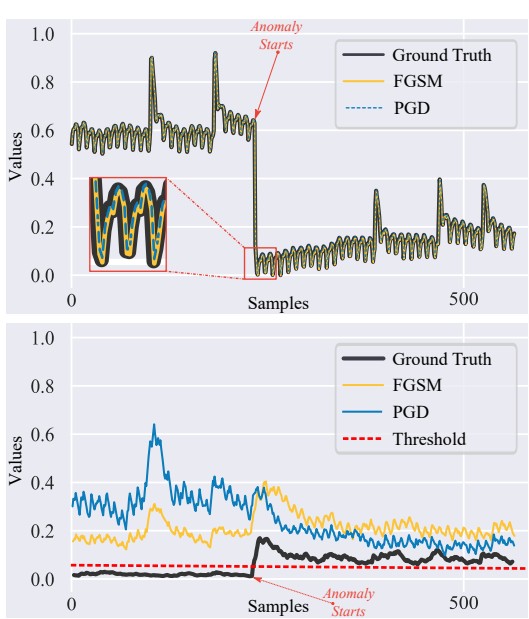

Figure 1: Example of ground truth and perturbed time series using FGSM and PGD attacks on CLMPPCA.

We present, transfer, and adapt adversarial attacks that have been demonstrated to work well on images to time series data (containing anomalies and intrusions) in this work. Additionally, we present an experimental study utilizing benchmark datasets from the aerospace and automobile industries and server machines, demonstrating that state-of-the-art anomaly and intrusion detection methods are vulnerable to adversarial attacks. We highlight specific real-world use cases to emphasize the critical nature of such attacks in real-world scenarios. Our findings indicate that deep networks for time series data, like their computer vision counterparts, are vulnerable to adversarial attacks. As a result, this paper emphasizes the importance of protecting against such attacks, particularly when anomaly and intrusion detection systems are used in sensitive industries such as aerospace and automobiles. Finally, we discuss some mechanisms for avoiding these attacks while strengthening the models' resistance to adversarial examples.

**Aim, Scope and Contribution.** In this work, we do not propose any novel adversarial attack method. However, we demonstrate the threat of existing attacks such as FGSM and PGD on state-of-the-art anomaly and intrusion detection methods. In comparison to the computer vision domain, where adversarial attack has been extensively studied and investigated, the literature on novelty detection, and particularly on anomaly detection, is noticeably devoid of such studies. The purpose of this paper is to bring attention to this issue. Additionally, we hope to encourage researchers to consider robustness to adversarial attacks when evaluating future detectors. The paper's scope was limited to SOTA anomaly detectors and intrusion detection systems (Note: As intrusion detection is a vast domain we consider only one sub-domain i.e., intrusion detection in Controller Area Network). Finally, to demonstrate that the current generation of detectors is unprepared against adversarial attacks. We demonstrate these attacks successfully on a deployed system in the aerospace industry.

## 2 RELATED WORK

### 2.1 BACKGROUND AND NOTATIONS

When performing a supervised learning task, we define $D = \{(s_i, y_i) | i = 1, \ldots, N\}$ to represent a data set containing N data samples. Each data sample is composed of a $m$-dimensional multivariate time series $s_i$ and a single target value $y_i$ for classification. We will observe such formation in intrusion detection scenarios (see Section 4.2). For unsupervised learning, each data sample is

again composed of a $m$-dimensional multivariate time series $s_i$ however, $y_i$ is an $n$-dimensional multivariate time series obtained from an autoregressive model, predicting the future. In most cases, $n = m$ however, they can be different as well. Moreover, we define any deep learning method as $\mathcal{F}(\cdot) \in f : \mathbb{R}^N \to \widehat{y}$ and loss function (e.g., cross entropy or mean squared error) as $\mathcal{L}_f(\cdot, \cdot)$. Finally, generating an adversarial instance $s_i^{adv}$ can be described as a optimization problem given a trained deep learning model $\mathcal{F}$ and an original input time series $s_i$, as follows:

$$\min \left\| s_i - s_i^{adv} \right\| \quad s.t. \quad \mathcal{F}(s_i) = \widehat{y_i}, \quad \mathcal{F}(s_i^{adv}) = \widehat{y_i^{adv}} \quad and \quad \widehat{y_i} \neq \widehat{y_i^{adv}}. \tag{1}$$

## 2.2 ADVERSARIAL ATTACKS

In 2014, Szegedy et al. (2013) introduced adversarial examples against deep neural networks for image recognition tasks for the first time. Following these inspiring discoveries, an enormous amount of research has been devoted to generating, understanding, and preventing adversarial attacks on deep neural networks (Eykholt et al., 2018; Goodfellow et al., 2014; Madry et al., 2017). Adversarial attacks can be broadly classified into two types: white-box and black-box attacks. As white-box attacks presume access to the model's design and parameters, they can attack the model effectively and efficiently using gradient information. By contrast, black-box attacks do not require access to the output probabilities or even the label, making them more practical in real-world situations. However, black-box attacks frequently take hundreds, if not millions, of model queries to calculate a single adversarial case.

The majority of adversarial attack techniques have been proposed for use in image recognition. For instance, a Fast Gradient Sign Method attack was developed by Goodfellow et al. (2014) as a substitute for expensive optimization techniques (Szegedy et al., 2013). Madry et al. (2017) proposed Projected Gradient Descent (PGD) in response to the success of FGSM. PGD seeks to find the perturbation that maximizes a model's loss on a particular input over a specified number of iterations while keeping the perturbation's size below a specified value called epsilon ($\epsilon$). This constraint is typically expressed as the perturbation's $L^2$ or $L^\infty$ norm. It is added to ensure that the content of the adversarial example is identical to that of the unperturbed sample — or even to ensure that the adversarial example is imperceptibly different from the unperturbed sample. Carlini-Wagner is another well-known attack (Carlini & Wagner, 2017). However, it is primarily intended for $L^2$ norm-based attacks, whereas this study focuses exclusively on $L^\infty$ norm-based attacks.

**Adversarial Attacks on Time Series.** Limited efforts have been made to extend similar attacks to time series data. Surprisingly, the community has ignored adversarial attack approaches for time series anomaly and intrusion detection tasks. However, a few adversarial attack approaches have been proposed recently for the time series classification task, which are tangentially related to our work. For instance, in their work on adopting a soft K Nearest Neighbors (KNN) classifier with Dynamic Time Warping (DTW), Oregi et al. (2018)demonstrated that adversarial examples could trick the proposed nearest neighbors classifier on a single simulated synthetic control dataset from the UCR archive (Dau et al., 2019). Given that the KNN classifier is no longer considered the state-of-the-art classifier for time series data (Bagnall et al., 2017), Fawaz et al. (2019) extend this work by examining the effect of adversarial attack on the more recent and commonly used ResNet classifier (He et al., 2016). Fawaz et al. (2019), on the other hand, focused mainly on univariate datasets from the UCR repository. As a result, Harford et al. (2020) investigate the influence of adversarial attacks on multivariate time series classification using the multivariate dataset from UEA repository (Bagnall et al., 2018). However, Harford et al. (2020) only consider basic methods such as 1-Nearest Neighbor Dynamic Time Warping (Seto et al., 2015) (1-NN DTW) and a Fully Convolutional Network (FCN). Karim et al. (2020) and Harford et al. (2020) attacked models using Gradient Adversarial Transformation Networks (GATNs). However, they examined just transfer attacks, a relatively weak sort of black-box attack. Only Siddiqui et al. (2019) demonstrated the effectiveness of gradient-based adversarial attacks on time series classification and regression networks. However, they considered a very simple baseline for the attack, containing only three convolutional, two max-pooling, and one dense layer.

Our study differs from previous research in that we focus on time series anomaly and intrusion detection rather than the broader classification problem. More precisely, we explore autoregressive models that have been mostly overlooked in prior works. Additionally, rather than targeting generic deep neural networks KNN with DTW or ResNet, we investigate state-of-the-art anomaly

Figure 2: Pipeline of a typical time series training phase and adversarial attack phase.

or intrusion detection methods. For instance, when it comes to anomaly detection, we focus on the most contemporary and commonly used techniques, such as MSCRED (Zhang et al., 2019), CLMPPCA Tariq et al. (2019), and MTAD-GAT Zhao et al. (2020). Similarly, for controller area network intrusion detection, we explore two well-known methods: CAN-ADF (Tariq et al., 2020a) and CANTransfer (Tariq et al., 2020b). Section 4 will cover these methods in further depth.

## 3 THREAT MODEL AND ATTACK GENERATION

**Adversary's Capabilities.** We consider an adversary whose objective is to reduce the effectiveness of a victim model. The attacker can apply the perturbations by modifying the victim's test-time samples, for example, by compromising a sensor or the data link that collects the data for inference. We investigate a $L^\infty$ norm threat model with a $0.1$ epsilon. Due to the variable input range of time series data, there are no box constraints, in contrast to the visual image, where the pixels take on a definite value between $[0, 255]$. As a result, the data was standardized in our case using a zero-mean and unit standard deviation which justified the choice of $0.1$ as the epsilon value.

**Adversary's Knowledge.** To evaluate the vulnerability of anomaly and intrusion detection systems, we examine non-targeted white-box scenarios in which the attacker has complete knowledge of the victim model, including its training data and the model's tunable parameters and weights.

**Adversary's Goals.** The adversary considers two cases: (i) normal to anomaly (or intrusion) and (ii) anomaly to normal. In (i), the adversary creates a $s_i^{adv}$ for each test sample $s_i$ so that the models interpret it as an anomaly (or intrusion), thereby generating a false-positive. However, in (ii), the adversary fabricates $s_i^{adv}$ to achieve the inverse effect, namely, to cause the model to predict an anomaly as normal, hence generating false-negative examples. As anomalies are rare events, even a few misclassifications caused by the adversary can have a detrimental effect on the model's performance.

**Adversarial Attack Generation.** The Fast Gradient Sign Method (FGSM) attack was proposed for the first time by Goodfellow et al. (2014). The training of neural networks entails minimizing a loss function by adjusting the network weights. FGSM, on the other hand, does the opposite by adjusting the input samples in the direction opposite to the loss function's minimum. Thus, the FGSM attack is concerned with the computation of optimal perturbation series $\eta$, which can be added/summed to an input sample pointwise (i.e., a point refers to a single timestep) in order to maximize the classification loss function, i.e., cause misclassifications. This is mathematically expressed as:

$$\eta = \epsilon \cdot \text{sign}\left(\nabla_s \, \mathcal{L}_f(s_i, y_i)\right) \tag{2}$$

where $\nabla_s$ denotes the derivative of the network's loss, $\mathcal{L}_f(\cdot, \cdot)$, with respect to each timestep in $s_i$ (calculated for an input datapoint $s_i$ and it's true output $y_i$). To control the magnitude of the perturbation (i.e., to keep it imperceptibly small), $\epsilon$ is used as a multiplier factor. After that, the perturbed sample $s_i^{adv}$ can be computed as $s_i + \eta$. Note that FGSM requires the attacker to compute the loss function gradient with respect to a given input, which may not be possible directly. Due to the fact that FGSM requires knowledge of the internal workings of the network, therefore referred to as a white-box attack. However, a surrogate model can be used to simulate the target model. An FGSM attack can be applied to the surrogate to generate adversarial examples (Papernot et al., 2017), allowing for the use of such white-box attacks in practical practice scenarios (Kurakin et al., 2016b).

Table 1: A summary of anomaly and intrusion detection datasets.

| Statistics | SMAP | MSL | SMD | KARI | Synthetic | OTIDS |
|---|---|---|---|---|---|---|
| Dimensions | 55 | 27 | 28 | 4-35 | 30 | 11 |
| Anomalies (or Intrusions) | 13.13% | 10.27% | 4.16% | 1% | 1.10% | 48.64% |
| Train Size | 135,183 | 58,317 | 708,405 | 4,405,636 | 8,000 | 2,306,954 |
| Test Size | 427,617 | 73,729 | 708,420 | 17,622,546 | 10,000 | 2,306,955 |

Madry et al. (2017) proposed a more robust adversarial attack called Projected Gradient Descent (PGD). This attack employs a multi-step procedure and a negative loss function. It overcomes the problem of network overfitting and the shortcomings of the FGSM attack. It is more robust than first-order network information-based FGSM, and it performs well under large-scale constraints. Gradient Descent is essentially identical to the Basic Iterative Method (BIM) (Kurakin et al., 2016b) or the Iterative FGSM (IFGSM) (Kurakin et al., 2016a) attacks. The only difference is that PGD initializes the example at a random location within the ball of interest (determined by the $L^\infty$ norm) and performs random restarts, whereas BIM initializes at the original location.

$$s_{i,t+1}^{adv} = \Pi_{s+\delta} \left( s_{i,t}^{adv} + \alpha \, \text{sign}(\nabla_s \mathcal{L}_f(s_i, y)) \right) \quad s.t. \quad 1 \le t \le T \tag{3}$$

where $\delta$ is a nonempty compact topological space, $T$ is the total number of iterations, and $\alpha$ is the control rate. An illustration of the overall pipeline is provided in Figure 2.

## 4 EXPERIMENTAL SETUP

**Datasets.** We employ a variety of datasets to detect anomalies and intrusions. For anomaly detection we employ three public datasets: (i) Mars Science Laboratory rover (MSL) (Hundman et al., 2018), (ii) Soil Moisture Active Passive satellite (SMAP) (Hundman et al., 2018), and (iii) Server Machine Dataset (SMD) (Su et al., 2019), as well as one private dataset: (vi) Korean Aerospace Research Institute KOMPSAT-5 satellite (KARI) (Tariq et al., 2019) and one synthetic dataset: (v) from the MSCRED paper (Zhang et al., 2019). Intrusion detection was performed using the CAN Dataset for intrusion detection (OTIDS) (Lee et al., 2017). The datasets were chosen based on our baselines' shown ability to provide state-of-the-art performance on these datasets. Table 1 summarize these datasets.

**Evaluation Metrics.** For intrusion detection, we use three metrics: Precision, Recall, and F1-score. However, we only provide the $F_1$ score in the main text (please refer to Appendix X for detailed results). To obtain the final classification result for anomaly detection methods, we observed that the majority of detectors use a thresholding method on top of the neural network's predictions, which are expressed as an anomaly score or prediction error. The prediction, recall, and F1-score are then calculated using the results from thresholding methods. While these metrics are beneficial, the true impact of the adversarial attack is visible primarily in anomaly detectors' anomaly score and prediction errors. Therefore, we include Figure 1, 3b and 3a, as illustrations of this impact. Additionally, we include more related figures in Appendix A– G.

We conduct experiments on the following baselines to demonstrate that the vulnerability to adversarial attacks is common among several state-of-the-art anomaly (or intrusion) detection architectures.

### 4.1 ANOMALY DETECTION BASELINES

Anomaly detectors based on Deep Neural Networks (DNNs) are the most frequently used method. However, some methods based on Graph Neural Networks (GNNs) have also been proposed recently. As a result, we evaluated both DNNs- and GNNs-based anomaly detectors. We used the following criteria to determine the baseline: (i) To ensure that we cover a broad range of methods, we decide that the baselines we choose should be diverse, i.e., no two baselines have similar model architecture. (ii) They should consider a different pre-processing technique (e.g., using raw data or feature vectors). (iii) They should take into account various post-processing techniques for prediction (e.g., anomaly score, prediction error, or classification score). (iv) The method is widely accepted and peer-reviewed. For this criterion, we take into account GitHub Forks, paper citations, and publication venues. (v) The source code is freely available or can be obtained upon request. We selected the following baselines based on these criteria:

- **MSCRED (Zhang et al., 2019) [AAAI].** Taking advantage of the temporal dependencies inherent in multivariate time series, Zhang et al. (2019) proposed a Multi-Scale Convolutional Recurrent Encoder-Decoder (MSCRED) for anomaly detection on two datasets: I synthetic and (ii) power plant. Sidenote: Shen et al. (2020) demonstrated that MSCRED outperforms all SOTA anomaly detection methods except Temporal Hierarchical One-Class (THOC), but we were unable to evaluate THOC as the code is not available (see more details below this list). As a result, we chose the second best method (i.e., MSCRED) among recently developed SOTA anomaly detection methods. Because the power plant dataset is not publicly available, we compare MSCRED with and without adversarial attack using the synthetic dataset used by Zhang et al. (2019) in their work.

- **CLMPPCA (Tariq et al., 2019) [KDD].** Tariq et al. (2019) proposed a hybrid approach for anomaly detection in multivariate satellite telemetry data. They propose a Convolutional LSTM with Mixtures of Probabilistic Principal Component Analyzers (CLMPPCA) method for transforming the time window containing several telemetry data samples into a feature vector that is used to train the model and to predict the future data instances. To make final classification, the prediction errors calculated from the prediction and ground truth are combined with a moving average-based threshold method. Tariq et al. (2019) evaluated a private dataset from the Korean Aerospace Research Institute's KOMPSAT-5 satellite (KARI). We were able to obtain the same private dataset and demonstrate how adversarial attacks affect the performance of CLMPPCA. One of the primary reasons for selecting CLMPPCA is that it is currently deployed at KARI. Thus, successfully demonstrating an attack on this method will demonstrate its applicability in a practical scenario.

- **MTAD-GAT (Zhao et al., 2020) [ICDM].** Zhao et al. (2020) proposed a multivariate time series anomaly detector based on Graph Attention Networks. The authors treat each univariate time series as a separate feature and employ two parallel graph attention layers to learn the complex dependencies between multivariate time series in both temporal and feature dimensions by jointly optimizing a forecasting-based and reconstruction-based model. MTAD-GAT outperformed several recent time series anomaly detectors such as OmniAnomaly (Su et al., 2019), MAD-GAN (Li et al., 2019), and DAGMM (Zong et al., 2018) from ICLR 2018, on three publicly available anomaly datasets (SMAP, MSL, and SMD). As a result, MTAD-GAT is one of the best SOTA methods currently available. We evaluate MTAD-GAT with and without adversarial attacks on all three datasets.

Note that we chose these three baselines based on their compliance with our defined criteria. Additionally, we were unable to evaluate some recent methods, such as Temporal Hierarchical One-Class (THOC) published at NeurIPS 2020, because the source code is not publicly available and our request to obtain the source code from the author was not answered. We discuss this further in Section 9.

## 4.2 INTRUSION DETECTION BASELINES

Khraisat et al. (2019) and Gamage & Samarabandu (2020) discuss the different domains where Intrusion detection-based methods are employed. We selected one such area, i.e., Intrusion detection in Vehicle Controller Area Network, and compared state-of-the-art solutions from this domain. We use the same criteria as anomaly detection to select the intrusion detection baselines as follows:

- **CAN-ADF.** Tariq et al. (2020a) proposed a hybrid of heuristics and recurrent neural network (RNN) to detect different intrusions such as DoS, Fuzzing, and Replay attacks. The heuristics and RNN components in CAN-ADF complement each other resulting in significantly better performance and stability. The gradient-based attack can only be applied to the RNN component of CAN-ADF. Therefore, we believe it to be an interesting scenario where only one of the two components of the detector is attacked.

- **CANTransfer.** Recently, continual learning-based methods have become prevalent for intrusion detection. They can learn new intrusion signatures with only a few samples while keeping the same performance on previously learned intrusions. CANTransfer (Tariq et al., 2020b) is one such method. The author first trains the base model on DoS Attack and then uses continual learning techniques to learn new intrusion such as Fuzzing and Replay by using only one attack

Table 2: MTAD-GAT results (F$_1$ score) on MSL, SMAP and SMD datasets.

| Method | MSL | SMAP | SMD |
|---|---|---|---|
| No Attack | 0.950 | 0.894 | 0.999 |
| FGSM | 0.719 | 0.804 | 0.803 |
| PGD | **0.687** | **0.775** | **0.665** |

Table 3: MSCRED results (F$_1$ score) on synthetic dataset from original paper.

| Method | Pre. | Rec. | F$_1$ |
|---|---|---|---|
| No Attack | 1.000 | 0.800 | 0.890 |
| FGSM | 0.487 | 0.500 | 0.493 |
| PGD | **0.485** | **0.500** | **0.492** |

instance. We believe that it will be interesting to observe the robustness of continual learning-based intrusion detectors against adversarial attacks.

Both CAN-ADF and CANTransfer use the same OTIDS dataset. Therefore, it gives us the opportunity to perform a fair comparison of two different systems (under adversarial attack) solving the same problem.

## 5 EMPIRICAL EVALUATION

We present results for the $L^\infty$ FGSM and PGD attacks against three SOTA anomaly detection methods—MSCRED, CLMPPCA, and MTAD-GAT—as well as two controller area network intrusion detection methods—CAN-ADF and CANTransfer. The Appendix includes additional details about the $L^\infty$ results (Appendix A); details on the $L^1$ (Appendix B) and $L^2$ (Appendix C) attacks; more details on impact of adversarial attacks on MTAD-GAT (Appendix D– F some original vs. perturbed time series samples (Appendix G; and results from the FGSM, PGD, BIM, Carlini-Wagner, and Momentum Iterative Method (MIM) (Dong et al., 2018) attacks on 71 datasets from the UCR repository (Appendix H). In general, we observe that perturbations that are $L^\infty$-bounded are more effective. This could be explained by optimization challenges, as $L^1$ and $L^2$ attacks are typically more difficult to optimize (Carlini & Wagner, 2017; Tramer & Boneh, 2019).

### 5.1 ADVERSARIAL ATTACK ON ANOMALY DETECTORS

**MSCRED Performance.** We employ non-targeted FGSM and PGD methods to attack MSCRED. As a result, only $s_i$ from the test set is made available to the attack methods. The $\epsilon$ is set to $0.1$ for the FGSM attack, and $\alpha$ is set to $0.1$ for the PGD attack with $T = 40$. The MSCRED method determines the appropriate threshold between normal and anomalous data points based on the training data. As a result, any modification to the test samples should not affect the threshold. As shown in Table 3, the victim model (MSCRED) has no efficacy on the samples perturbed by FGSM and PGD attacks and thus fails to detect all anomalies. Additionally, MSCRED classifies all instances of normal data as anomalies. We demonstrate in Figure 3a that MSCRED (No Attack) can accurately predict the majority of anomalies with an F$_1$ score of $0.890$ (see see Table 3). According to Figure 3a, the anomaly scores under FGSM (yellow) and PGD (blue) attacks are always higher than the threshold (red dashed line), which means that MSCRED is predicting everything as an anomaly, resulting in an F$_1$ score of less than $0.50$. It is intriguing that such a small amount of change in the time series, which is primarily imperceptible to the naked eye, can greatly affect the MSCRED's anomaly scores, even when the perturbations are so minute. The results in Table 3 are demonstrating that MSCRED is not robust against adversarial attacks.

**MTAD-GAT Performance.** As with MSCRED, we attack MTAD-GAT using a non-targeted FGSM and PGD method with $\epsilon = 0.1$, $\alpha = 0.1$, and $t = 40$. The results of adversarial attacks against MTAD-GAT trained on the MSL, SMAP, and SMD datasets are shown in Table 2. MTAD-GAT demonstrates state-of-the-art performance for anomaly detection in the absence of an adversarial attack (No Attack). However, when adversarial examples from FGSM and PGD are used to evaluate it, the detection performance drops to as low as 66%. The impact of a PGD attack is more significant than that of the FGSM attack, which is understandable given that PGD is a more powerful attack than FGSM. It leads us to ponder that if more sophisticated attacks are explicitly developed for time series data, they will have a significantly greater impact on SOTA anomaly detectors. As a result, future anomaly detection methods should take adversarial examples into account.

Additionally, Figure 3b illustrates the effect of adversarial examples from FGSM (yellow) and PGD (blue) attacks on the MTAD-GAT anomaly score for the MSL dataset. We can see that the anomaly

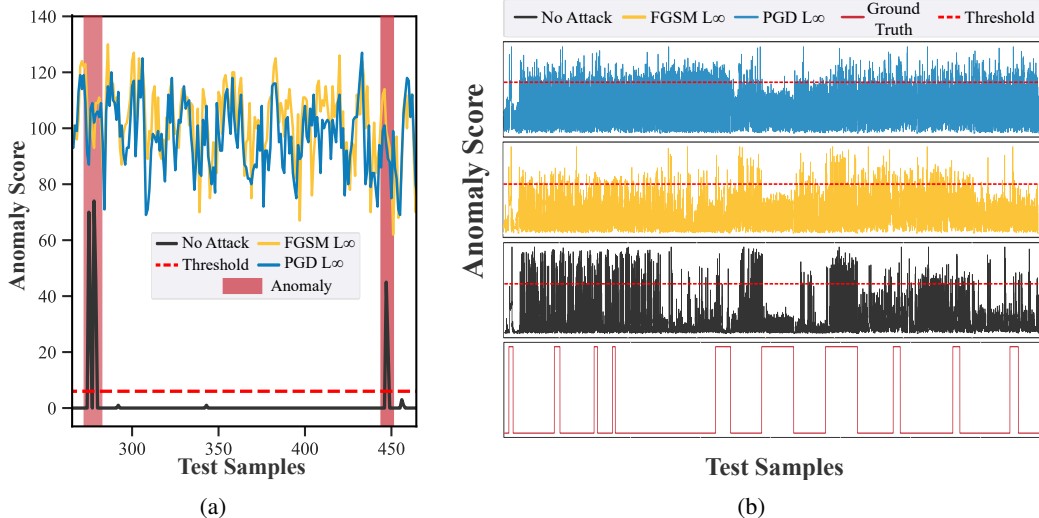

Figure 3: Anomaly score of No Attack, FGSM and PGD on MSCRED (a) and on MTAD-GAT for MSL dataset (b) The y-axis scale is between 0 and 1 for (b). See Appendix A and D for more details on (a) and (b), respectively.

Table 4: CLMPPCA prediction errors on KARI KOMPSAT-5 dataset for subsystems (SS) 1-10.

| Methods | SS1 | SS2 | SS3 | SS4 | SS5 | SS6 | SS7 | SS8 | SS9 | SS10 |
|---|---|---|---|---|---|---|---|---|---|---|
| No Attack | 0.025 | 0.020 | 0.646 | 0.018 | 0.078 | 0.081 | 0.028 | 0.015 | 0.043 | 0.106 |
| FGSM | 0.306 | 0.327 | 5.657 | 0.153 | 1.744 | 1.708 | 0.246 | 0.201 | 1.303 | 0.314 |
| PGD | **0.688** | **0.748** | **11.200** | **0.205** | **2.459** | **3.391** | **0.430** | **0.231** | **1.798** | **0.555** |

score for FGSM and PGD frequently exceeds the threshold (red dashed line), resulting in a large number of false positives and lowering the $F_1$ score from 94.98% to 71.90% for FGSM and 68.69% for PGD.

**CLMPPCA Performance.** The KARI dataset is divided into ten subsystems. As a result, we trained the CLMPPCA model on each subsystem separately, as described in the original paper. We then used FGSM and PGD attacks to evaluate each of these trained models. For FGSM, we use $\epsilon = 0.1$, for PGD, we use $\alpha = 0.1$, and $t = 40$. Table 4 summarizes the prediction errors for each subsystem prior to and following the attack. We can see that when adversarial attacks are used, the prediction error increases up to twentyfold. Note: For brevity and space constraints, we omit the $F_1$ score from Table 4, as it is $0.50$ for all subsystems. CLMPPCA fails to predict any anomalies under FGSM and PGD attacks because the prediction error is always higher than the threshold (see Figure 1). We believe that by employing these straightforward yet effective attacks, an adversary can easily introduce false positives into CLMPPCA's predictions at will, posing significant difficulties for satellite operators.

Our findings indicate that the majority of SOTA anomaly detectors prioritized performance over robustness. This could have dire consequences if such systems are deployed in real-world systems. CLMPPCA is one such example, which is currently being deployed at KARI. Please note that we have informed KARI of the vulnerability in CLMPPCA; additional information is available in our Ethics Statement (see Section 8).

## 5.2 ADVERSARIAL ATTACK ON CAN INTRUSION DETECTORS

We evaluate the robustness of two popular and recent Controller Area Network intrusion detectors under adversarial impact. We use the same experimental settings (i.e., same training and test sets, $epochs = 50$) and attack settings (i.e., $\epsilon = 0.1$, $\alpha = 0.1$ and $T = 40$) for CAN-ADF and CANTransfer to have a fair comparison.

**CAN-ADF Performance.** The results for CAN-ADF under no attack, FGSM attack, and PGD attack are presented in Table 5, column 2. While CAN-ADF outperforms CANTransfer in a normal scenario (i.e., no attack), it is also more vulnerable to adversarial attacks, with an $F_1$ score of $0.00$ for the PGD attack. Even though we could only attack the RNN component of CAN-ADF and the heuristics component remained unaffected, CAN-ADF's poor performance against adversarial attacks demonstrates its high reliance on the RNN component.

**CANTransfer Performance.** Table 5, column 3 contains the results for CANTransfer with no attack, FGSM attack, and PGD attack. While CANTransfer performs slightly worse than CAN-ADF in the absence of an attack ($0.879$), it exhibits greater resistance to adversarial attacks than CAN-ADF, particularly against FGSM attacks, where CANTransfer's performance drops by only $2\%$. We attempted multiple runs of the experiment, but the results remained

Table 5: CAN-ADF and CANTransfer results ($F_1$ score) on OTIDS dataset.

| Method | CAN-ADF | CANTransfer |
|---------|---------|-------------|
| No Attack | 0.987 | 0.879 |
| FGSM | 0.188 | 0.858 |
| PGD | **0.000** | **0.311** |

consistent for the FGSM attack. Under PGD, it is a different tale, as CANTransfer's performance drops significantly from $0.879$ to $0.311$, demonstrating its vulnerability.

These findings demonstrate that neither of the two most recent intrusion detectors for Controller Area Networks is resistant to adversarial examples. As a result, future research should consider the robustness of new detectors against adversarial attacks when designing them.

## 6 DISCUSSION

**Defense against Adversarial Time Series.** Adversarial training is one of the most commonly used defense methods against adversarial examples. However, as Kang et al. (2019) suggest, training a network to withstand one type of attack may weaken it against others. Additionally, Tramer et al. (2020) outline various methods to conduct an adaptive attack and demonstrate that none of the 13 recently developed defense methods can withstand all types of adaptive attacks. Recently, several techniques for defending against adversarial time series have been proposed. For example, Goodge et al. (2020) propose an Approximate Projection Autoencoder (APAE) resistant to IFGSM attacks. However, it only considers autoencoder-based anomaly detectors. Moreover, the performance of several SOTA baselines reported in the paper is significantly lower than that reported in their original paper using the same publicly available benchmark dataset. As a result, a thorough examination of the defense methods is required.

**Limitations and Future Work.** There are some limitations to our work, and future work will try to solve them. For instance, we could not evaluate all of the recent anomaly and intrusion detectors in our work due to the following reasons: (i) The most important reason is that the codes are not publicly available in many cases or the code is outdated, making it hard to compare ( we discuss this in detail in reproducibility section). (ii) It is hard to reproduce the same results as demonstrated by the paper, mainly when the codes are not from the original authors but developed by the community. Therefore, future work should look for more methods. Moreover, we have only applied FGSM, PGD, and SL1D (see Appendix) attacks on the detectors. We do provide results from other attacks such as Carlini-Wagner L2 and MIM on the UCR dataset in Appendix. Another future work will be to transfer these and new adversarial attacks to anomaly and intrusion detectors. Finally, developing robust detectors should be considered in future studies.

## 7 CONCLUSION

The concept of adversarial attacks on deep learning models for time series anomaly and intrusion detection was introduced in this paper. We defined and adapted adversarial attacks initially proposed for image recognition to time series data. On several benchmark datasets, we demonstrated how adversarial perturbations could reduce the accuracy of state-of-the-art anomaly and intrusion detectors. As data scientists and developers increasingly implement deep neural network-based solutions for time series related real-world critical decision-making systems (e.g., in aerospace and automobile industries), we shed light on several critical use cases where adversarial attacks could have severe and dangerous repercussions.

## 8 ETHICS STATEMENT

Our study, in our opinion, raises only one significant ethical issue (i.e., presenting the vulnerabilities of a deployed system). Now, we will describe how we deal with it. To begin, we downloaded the CLMPPCA code from GitHub. Second, we contacted the authors of the CLMPPCA paper and requested the dataset. Following KARI's security clearance. We were able to obtain access to the dataset and some code associated with the driver, which was kept private on purpose. We contacted the authors and informed them of our findings after identifying the vulnerabilities in CLMPPCA. The authors replicated our findings on the deployed system using the same attacks. For the time being, the system is offline, and the authors of the CLMPPCA paper and other KARI developers are investigating possible defense methods. We believe that adhering to this entire procedure resolves any ethical concerns regarding this matter.

## 9 REPRODUCIBILITY

**Reproducibility Issues in Baselines.** According to our research, the majority of recent anomaly detection methods do not make their source code publicly available. Additionally, many methods whose source code was made publicly available by their authors (or implemented unofficially) were outdated. As a result, we were unable to run them directly on the most recent machines. For instance, in our experiment, we used an Nvidia RTX 3090 GPU. We discovered that, due to some issues with the CUDA version, we could not run an older version of TensorFlow optimally. As a result, the code either takes an eternity to execute or does not execute at all.

**Our Solution.** We chose to port the baselines to the latest versions of TensorFlow and PyTorch, respectively, which were 2.5.0 for TensorFlow and 1.9.0 for PyTorch at the time of our experiments. We used the CleverHans Library's (Papernot et al., 2018) FGSM, PGD, BIM, Carlini-Wagner L2, SL1D, and MIM attacks , which were recently ported to TensorFlow2 and PyTorch in version 4.0.0. As a result, our workflows are compatible with the latest libraries. Additionally, after cleaning the code, we will include some tutorial attacks (similar to those included in the CleverHans library for image datasets) that can be used to assess the vulnerability of future detectors to adversarial attacks.

**Guidelines for Baseline:** Note that it is difficult to port or implement all of the most recent methods on our own. Therefore, we tried our best with the limited resources that we had to make the baselines compatible with the latest version of libraries. We will provide some guidelines for creating new baselines and evaluating them against adversarial attacks on our GitHub page. We will leave it up to the community to add additional methods in the future.

**Reproducibility of Evaluation Code:** We are currently cleaning up the code, and an initial draft of our repository is available at the following link: `https://anonymous.4open.science/r/ICLR298`. We will continue to update it. We request and welcome the reviewer to visit the repository and offer constructive feedback on improving it.

**Links to Baselines and Datasets.** We will include the updated codes for each baseline in our repository as well. We obtained the code of the baselines from the following repositories:

- MSCRED: `https://github.com/Zhang-Zhi-Jie/Pytorch-MSCRED`
- MTAD-GAT: `https://github.com/ML4ITS/mtad-gat-pytorch`
- CLMPPCA: `https://github.com/shahroztariq/CL-MPPCA`
- CAN-ADF: `https://github.com/shahroztariq/CAN-ADF`
- CANTransfer: `https://github.com/shahroztariq/CANTransfer`

Note that we are unable to share the KARI dataset as it is proprietary and requires security clearance to access. The link to rest of the dataset used in our evaluation are as follows:

- SMAP and MSL: `https://s3-us-west-2.amazonaws.com/telemanom/data.zip`
- SMD: `https://github.com/ML4ITS/mtad-gat-pytorch/tree/main/datasets`
- Synthetic: `https://github.com/Zhang-Zhi-Jie/Pytorch-MSCRED`
- OTIDS: `https://ocslab.hksecurity.net/Dataset/CAN-intrusion-dataset`

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

# A    DETAILS ON $L^\infty$ FGSM AND PGD ATTACKS

In Figure 4, we detail MSCRED's performance against $normmax$-norm FGSM and PGD attacks. Under normal conditions, we can see that the model correctly predicted three large anomalies but missed two minor ones. As a result, an $F_1$ score of $0.890$ is obtained. However, when attacked with either FGSM or PGD, the MSCRED model produces no meaningful results because it predicts everything as an anomaly. Furthermore, the patterns of anomaly score under FGSM and PGD attack are very similar to those observed during non-anomalous (or normal) periods. As a result, adjusting the threshold to account for changes in the anomaly score will not be as effective.

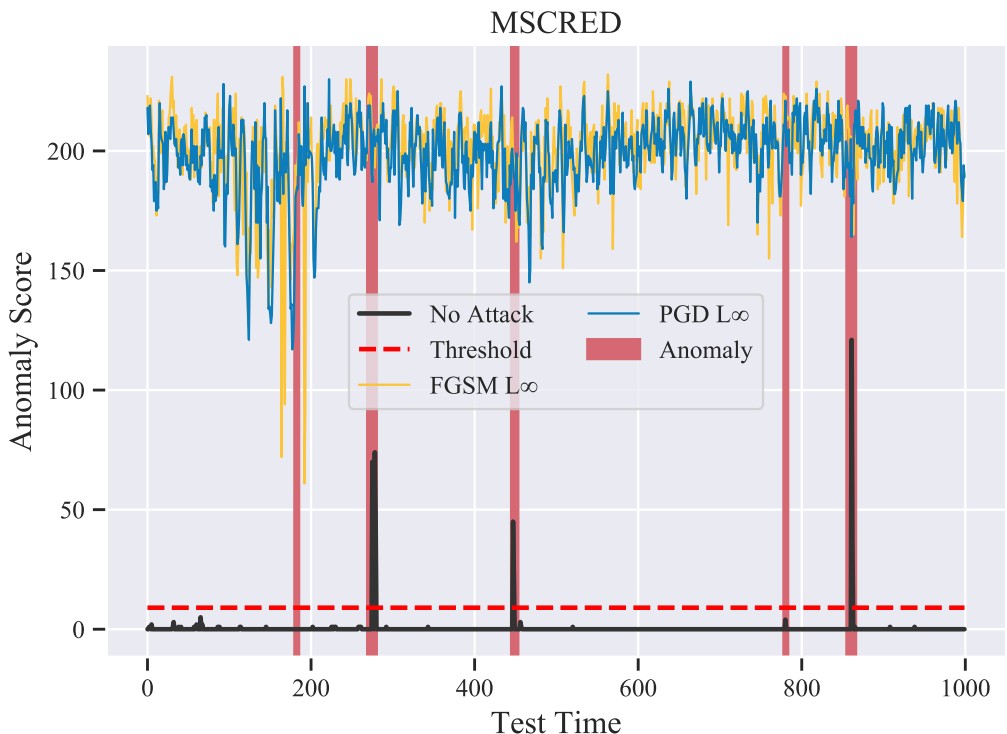

Figure 4: Anomaly score comparison of MSCRED under No Attack and, $L^\infty$-norm based FGSM and PGD attacks.

# B    SL1D AND FGSM $L^1$ ATTACK

In Figure 5, we present the results from two $L^1$ attacks: (i) FGSM $L^1$ and (ii) Sparse $L^1$ Descent (SL1D) attacks. As discussed previously in the main paper, optimizing $L^1$ and $L^2$-based attacks can be challenging. We can see an excellent illustration of this with the FGSM $L^1$ attack, where adversarial examples from the $L^1$-based FGSM attack produce nearly identical results to the No Attack data samples (with a few minor differences). However, the SL1D attack, also an $L^1$-based attack, performs similar to the $L^\infty$ attack discussed previously. Although the range of anomaly scores produced by SL1D attacks is slightly less than that produced by $L^\infty$ attacks, it is still significantly higher than the threshold making the MSCRED model to predict the whole input time series as an anomaly.

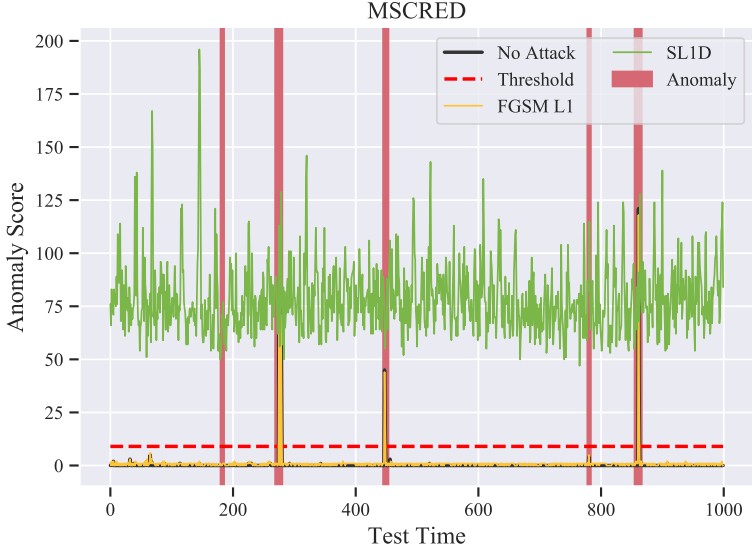

Figure 5: Anomaly score comparison of MSCRED under No Attack and, $L^1$-norm based FGSM and SL1D attacks.

## C  $L^2$ FGSM AND PGD ATTACK

The results of the $L^2$-based FGSM and PGD attacks are shown in Figure 6. Almost identical to the $L^1$-based FGSM attack, the $L^2$-based FGSM attack produces adversarial samples that have no effect on the anomaly score and are thus deemed ineffective. Similar results are obtained using the $L^2$-based PGD attack. As illustrated in Figure 6, the Anomaly scores for No Attack, FGSM $L^2$, and PGD $L^2$ all overlap significantly.

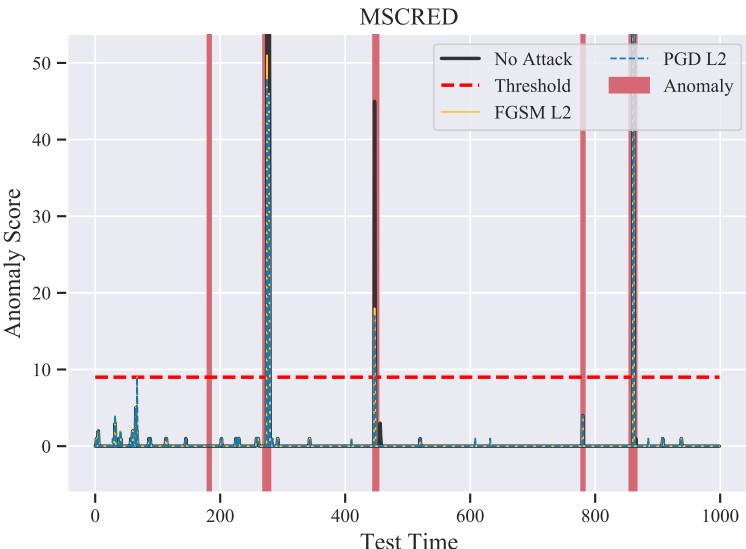

Figure 6: Anomaly score comparison of MSCRED under No Attack and, $L^2$-norm based FGSM and PGD attacks.

# D DETAILED VIEW OF MTAD-GAT RESULTS ON MSL DATASET

In this section, we discuss the MTAD-GAT results on the MSL dataset in greater detail. Figure 7– 9 show No Attack, FGSM attack, and PGD attack results on the entire test data, respectively. We can see that MTAD-GAT predicts fewer anomalies under FGSM and PGD attacks than normal conditions (i.e., No Attack), resulting in a higher rate of false negatives. We have now discussed both of these scenarios in detail in this work: (i) adversarial attack to generate false positives and (ii) adversarial attack to generate false negatives. Additionally, consistent with our previous findings, PGD performs better than FGSM and generates more false negatives than FGSM.

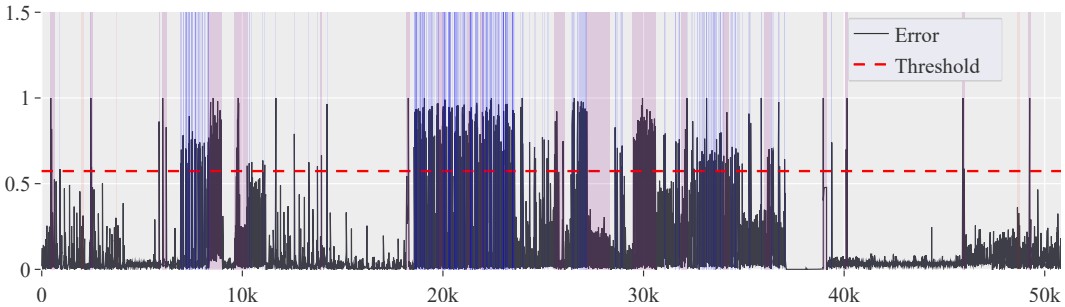

Figure 7: MTAD-GAT's anomaly score on normal data (i.e., No Attack).

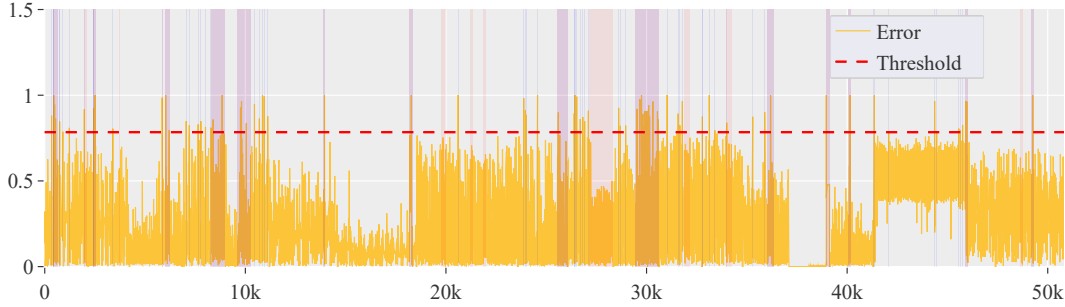

Figure 8: MTAD-GAT's anomaly score under FGSM attack.

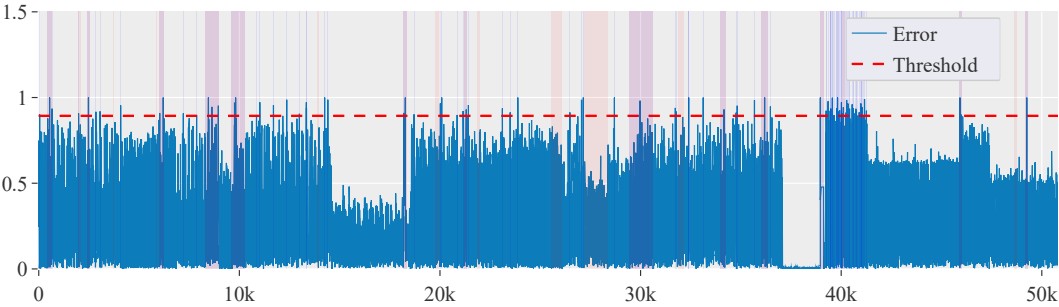

Figure 9: MTAD-GAT's anomaly score under PGD attack.

# E RESULTS ON SMD DATASET FOR MTAD-GAT

We present additional details on the MTAD-GAT results using the Server Machine Dataset (SMD) in Figure 10, 11a and 11b . In the figures, the top row (in red) represents the Anomaly scores, the middle row (in brown) represents the MTAD-GAT predictions, and the bottom row (in blue) represents the ground truth. We can see that MTAD-GAT performs at a state-of-the-art level under normal conditions. However, when subjected to FGSM and PGD attacks, it generates a large number of false positives, resulting in a significant decrease in overall performance. Additionally, we can observe that when PGD is used, MTAD-GAT produces more false positives than when FGSM is used.

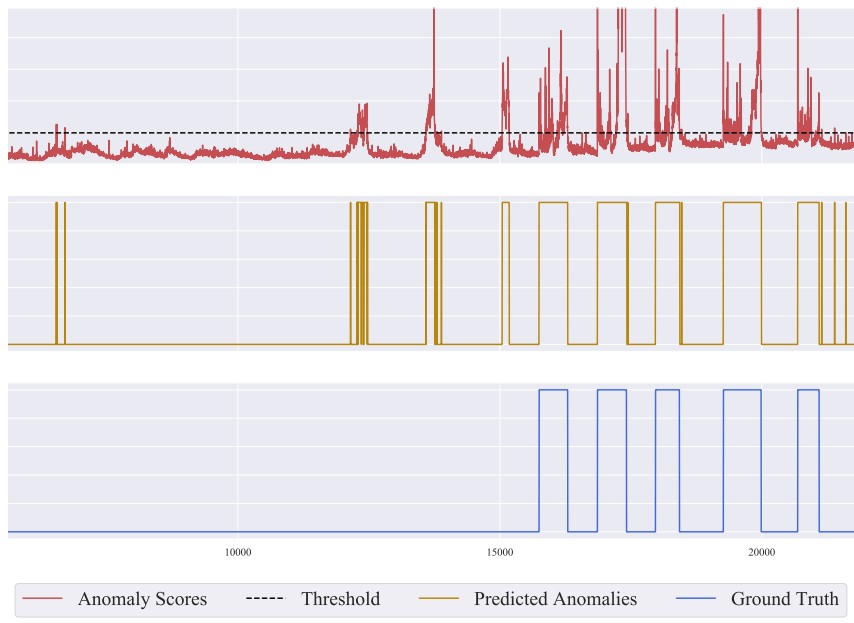

Figure 10: The anomaly score and predicted anomalies for MTAD-GAT on SMD dataset under normal conditions (No Attack).

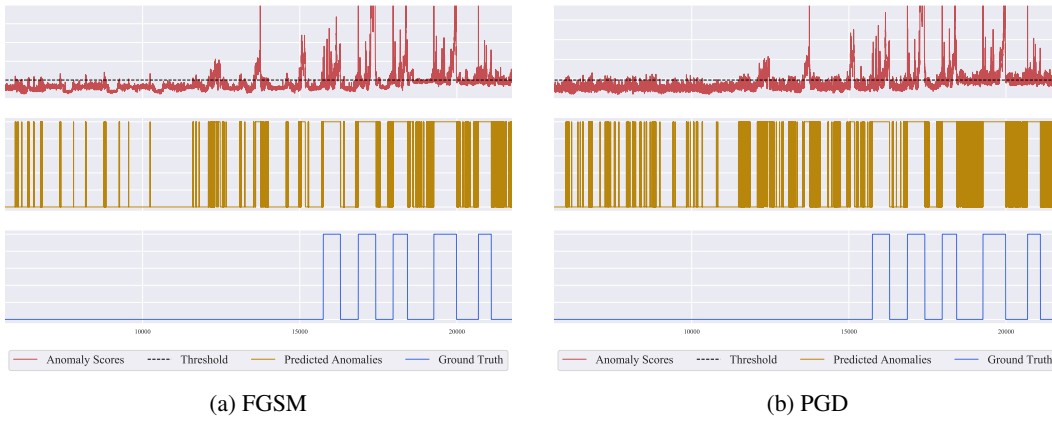

(a) FGSM

(b) PGD

Figure 11: The anomaly score and predicted anomalies for MTAD-GAT on SMD dataset under FGSM and PGD Attack.

## F EFFECTS OF FGSM AND PGD ATTACKS ON MTAD-GAT'S FEATURES

As previously stated, MTAD-GAT is composed of two components (i.e., forecasting and reconstruction). We demonstrate in Figure 12– 14 that both components become equally effective when subjected to adversarial attacks. For example, in normal circumstances (as illustrated in Figure 12), the forecast and reconstruction are quite close to the $y_i$ (ground truth). However, when attacked by FGSM, they deviate from the ground truth, fooling the system into believing it is an anomaly. Additionally, forecast and reconstruction are more chaotic during a PGD attack. As a result, detection performance is even lower than that of a FGSM attack.

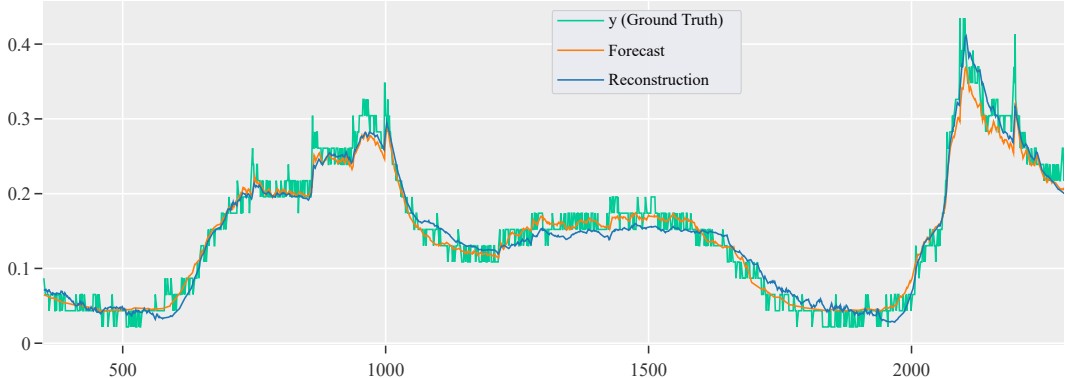

Figure 12: Comparison of Forecast and Reconstruction with $y_i$ during No Attack.

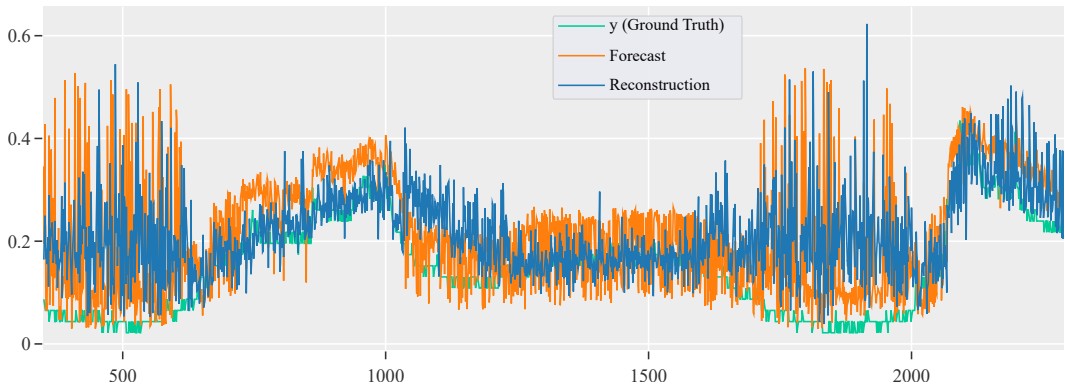

Figure 13: Comparison of Forecast and Reconstruction with $y_i$ during FGSM Attack.

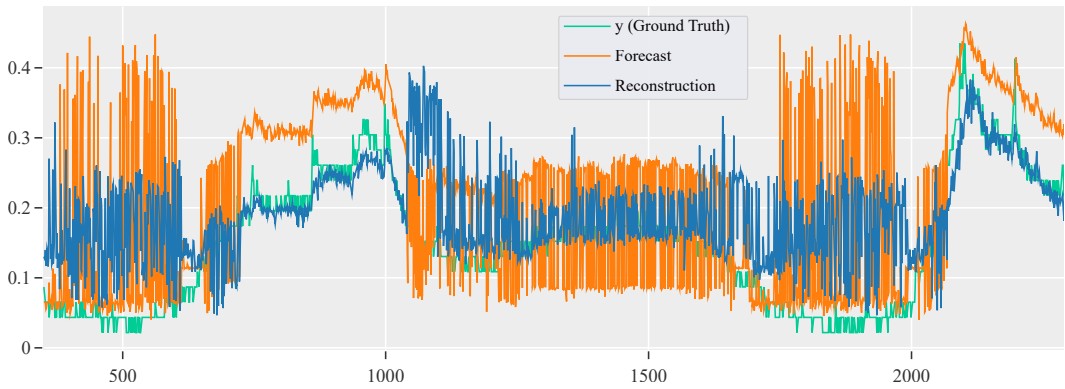

Figure 14: Comparison of Forecast and Reconstruction with $y_i$ during PGD Attack.

## G    ORIGINAL VS. PERTURBED SAMPLES

We compare some samples of original and perturbed time series in this section. The ground truth (in black), the FGSM (in yellow), and the PGD are depicted in Figure 15. (in dotted blue line). We can easily see that all three of the time series overlap, rendering them largely indistinguishable to the naked eye. Additionally, Figure 16a– 16c show an expanded version of the time series depicted in Figure 1. Each of the three time series (i.e., No Attack, FGSM, and PGD) appears identical. Here, we demonstrate that even simpler adversarial attacks such as FGSM and PGD can be highly effective on time series data. Such perturbations will go unnoticed by a human observer.

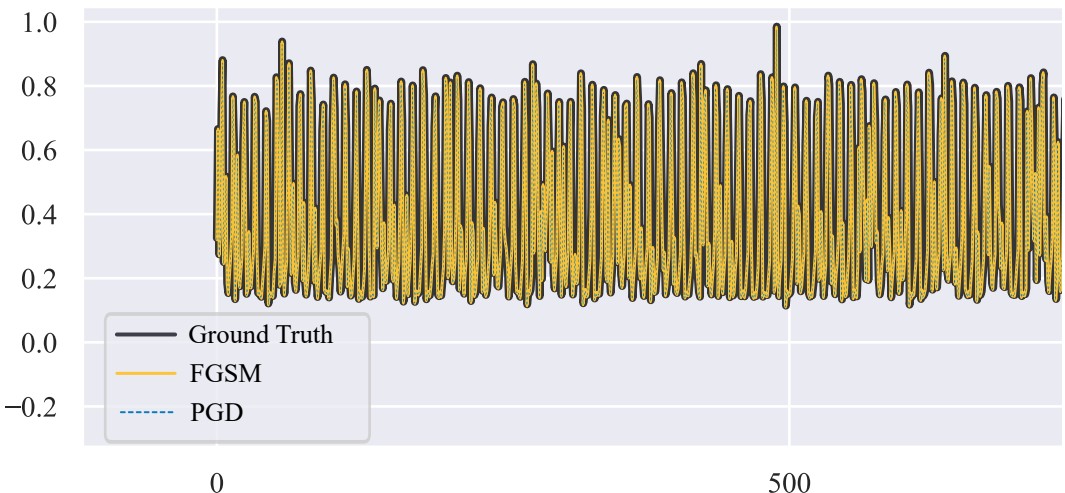

Figure 15: Comparing No Attack (Ground Truth), FGSM and PGD attack on time series from the KARI Subsystem 2 (SS2). There is no significant difference between the three time series therefore they seem perfectly overlap.

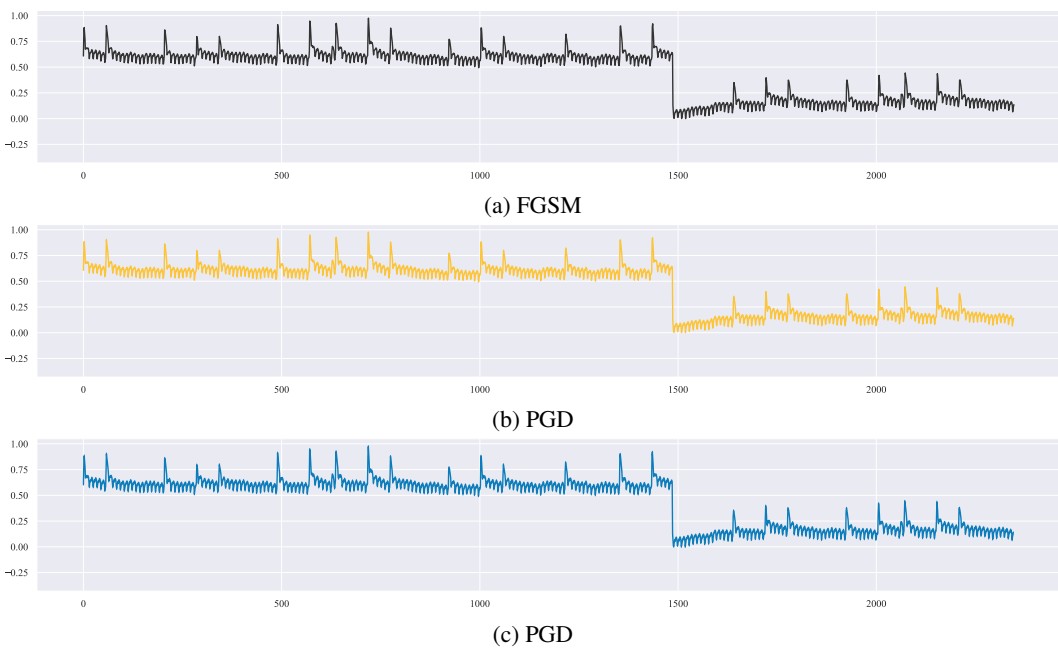

(a) FGSM

(b) PGD

(c) PGD

Figure 16: A more detailed view of the same time series as in Figure 1. However, for comparison purposes, the time series generated by No Attack (a), FGSM (b), and PGD (c) are shown separately.

# H   UCR DATASET RESULTS

In addition to all the experiment on state-of-the-art anomaly and intrusion detection system. We also cover general time series classification task where we attack a multilayer perception (MLP), a fully convolutional network and ResNet trained on different dataset from the UCR repository. We conduct an analysis of 71 datasets from the University of California, Riverside (UCR) repository. In future work, we will expand on this experiment by including additional neural networks (MobileNet, EfficientNet, DenseNet, and Inception Time) and datasets (the remainder of the UCR dataset, datasets from the UEA repository).

We find that the Carlini-Wagner $L^2$ attack provides the best adversarial examples, resulting in the most significant performance degradation. In Figure 17, we show some original samples and the corresponding perturbed samples generated by FGSM, PGD, BIM, Carlini-Wagner $L^2$, and MIM attacks on UCR datasets. Additionally, we present the ResNet classification results in Figure 17. Finally, in Table 6– 8, we present the classification results for MLP, FCN, and ResNet.

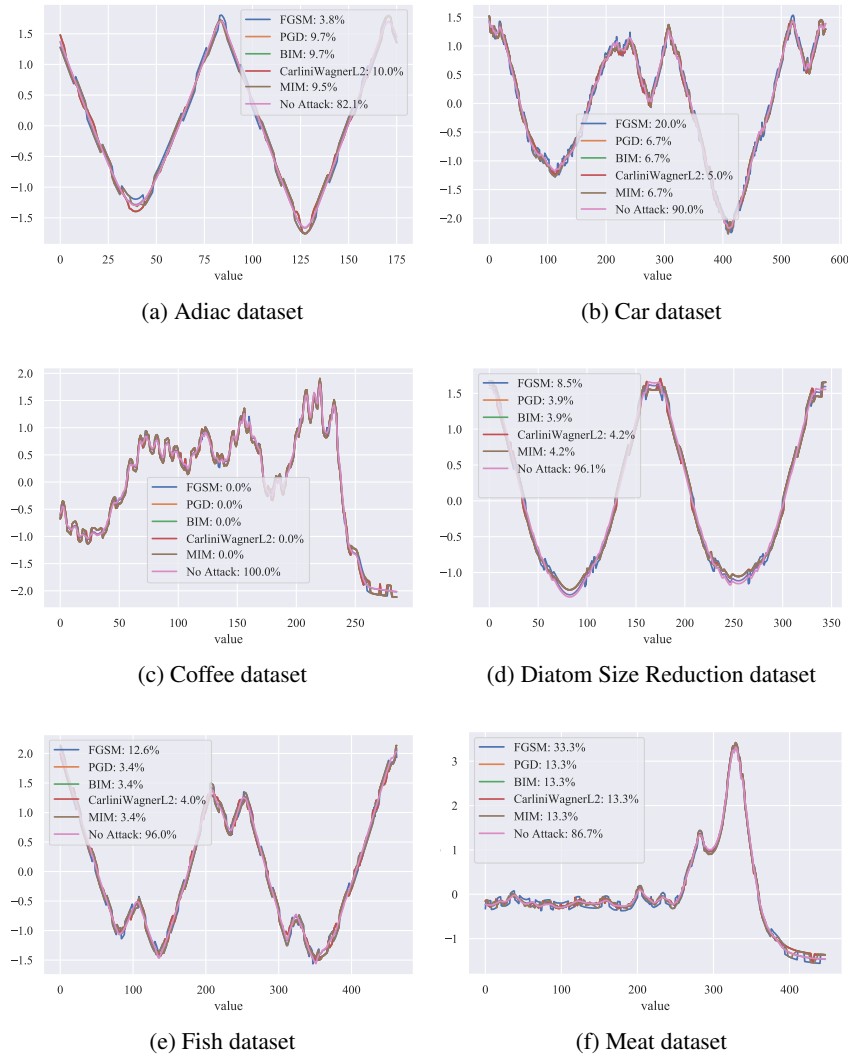

(a) Adiac dataset

(b) Car dataset

(c) Coffee dataset

(d) Diatom Size Reduction dataset

(e) Fish dataset

(f) Meat dataset

Figure 17: Comparison of original vs. perturbed samples from different datasets inside UCR repository. Moreover, the classification results of ResNet under normal and five different attack scenarios is also provided.

Table 6: Multilayer perceptron (MLP) classification result on UCR repository under five adversarial attacks.

| MLP | | | | | | |
|---|---|---|---|---|---|---|
| **Datasets** | **FGSM** | **PGD** | **BIM** | **Carlini Wagner** $L^2$ | **MIM** | *No Attack* |
| 50words | 44±0.8 | 42±1.3 | 42±1.3 | 35±1 | 43±1 | *63±1.1* |
| Adiac | 14±1.8 | 15±1.6 | 15±1.6 | 16±1.3 | 16±1.8 | *53±2.7* |
| ArrowHead | 29±3.9 | 27±3.3 | 27±3.3 | 24±4.3 | 27±3.1 | *74±2.6* |
| Beef | 32±5.1 | 26±3.9 | 26±3.9 | 29±3.9 | 27±3.4 | *78±3.9* |
| BeetleFly | 74±7.7 | 74±7.7 | 74±7.7 | 70±5 | 74±7.7 | *75±13.3* |
| BirdChicken | 62±5.8 | 62±5.8 | 62±5.8 | 57±10.5 | 62±5.8 | *69±2.9* |
| Car | 49±1 | 35±2.9 | 35±2.9 | 52±1 | 45±1 | *83±1* |
| CBF | 76±2.6 | 76±2.4 | 76±2.4 | 63±3.5 | 76±2.6 | *94±2.6* |
| Chlorine Concentration | 24±0.3 | 24±0.5 | 24±0.5 | 24±0.7 | 24±0.4 | *65±0.4* |
| Coffee | 9±2.1 | 9±2.1 | 9±2.1 | 9±4.2 | 9±2.1 | *100±0* |
| Computers | 46±1.1 | 45±1.1 | 45±1.1 | 45±1.1 | 45±1.1 | *58±0.9* |
| Cricket_X | 26±0.7 | 25±0.7 | 25±0.7 | 21±1 | 26±0.9 | *45±1* |
| Cricket_Y | 30±0.8 | 29±1.7 | 29±1.7 | 24±0.6 | 29±1.7 | *48±1.6* |
| Cricket_Z | 32±0.7 | 30±1.1 | 30±1.1 | 25±1 | 31±0.2 | *44±1.2* |
| DiatomSize Reduction | 40±1.2 | 37±1.4 | 37±1.4 | 31±4 | 38±1.5 | *95±2.4* |
| DistalPhalanx OutlineAgeGroup | 16±0.9 | 16±1 | 16±1 | 16±1 | 16±0.9 | *83±0.8* |
| DistalPhalanx OutlineCorrect | 29±1.4 | 28±1.8 | 28±1.8 | 25±0.9 | 29±1.7 | *77±0.9* |
| Distal PhalanxTW | 13±0.8 | 12±1.1 | 12±1.1 | 12±0.9 | 12±1.2 | *78±0.7* |
| Earthquakes | 69±1.5 | 69±1.5 | 69±1.5 | 52±4.2 | 69±1.5 | *73±1.1* |
| ECG200 | 60±1.8 | 60±2.1 | 60±2.1 | 29±5.8 | 60±2.1 | *84±0.6* |
| ECG5000 | 65±0.2 | 64±0.3 | 64±0.3 | 61±0.3 | 64±0.4 | *93±0.2* |
| ECGFiveDays | 48±2.3 | 46±2.1 | 46±2.1 | 35±4.8 | 47±2.1 | *95±3.3* |
| ElectricDevices | 22±0.4 | 21±0.5 | 21±0.5 | 21±0.6 | 21±0.6 | *55±0.8* |
| FaceAll | 57±0.3 | 56±0.4 | 56±0.4 | 39±0.9 | 56±0.2 | *74±0.6* |
| FaceFour | 79±2.4 | 77±2 | 77±2 | 76±1.8 | 79±1.4 | *88±0.7* |
| FacesUCR | 67±1.7 | 63±1.6 | 63±1.6 | 55±1.6 | 65±1.8 | *83±1.2* |
| FISH | 16±2.1 | 8±1.2 | 8±1.2 | 14±1.9 | 12±1.2 | *85±0.4* |
| Gun_Point | 48±6.1 | 47±6.2 | 47±6.2 | 34±5.4 | 47±6.2 | *92±1.4* |
| Ham | 34±2.4 | 34±2.6 | 34±2.6 | 48±3.5 | 34±2.6 | *70±2* |
| Haptics | 21±0.9 | 21±0.8 | 21±0.8 | 21±1.2 | 20±0.7 | *41±0.7* |
| Herring | 50±1.9 | 50±1.9 | 50±1.9 | 50±1.9 | 50±1.9 | *51±1.9* |
| InlineSkate | 21±1.1 | 19±0.8 | 19±0.8 | 20±1.4 | 20±1.4 | *34±0.7* |
| InsectWing beatSound | 37±0.7 | 30±0.3 | 30±0.3 | 42±0.3 | 34±0.4 | *62±0.7* |
| ItalyPower Demand | 82±0.8 | 82±0.9 | 82±0.9 | 11±1.4 | 82±0.9 | *96±0.2* |
| LargeKitchen Appliances | 33±2.2 | 32±1.3 | 32±1.3 | 34±0.6 | 33±2.1 | *51±0.5* |
| Lighting2 | 70±2.6 | 70±2.6 | 70±2.6 | 58±3.8 | 70±2.6 | *65±3.5* |
| Lighting7 | 53±4.2 | 53±3.7 | 53±3.7 | 35±3.7 | 53±3.7 | *64±2.4* |
| Meat | 26±1 | 26±1 | 26±1 | 25±1.7 | 26±1 | *74±1* |
| MedicalImages | 39±1.9 | 36±2.2 | 36±2.2 | 26±0.5 | 37±2.2 | *67±0.5* |
| MiddlePhalanx OutlineAgeGroup | 32±10.7 | 26±4.8 | 26±4.8 | 20±0.8 | 27±5.7 | *73±1.5* |

| | | | | | | |
|---|---|---|---|---|---|---|
| MiddlePhalanx OutlineCorrect | 46±1.5 | 46±1.6 | 46±1.6 | 45±1.5 | 46±1.6 | *56±1.5* |
| Middle PhalanxTW | 18±2.9 | 18±2.8 | 18±2.8 | 18±1.7 | 18±2.9 | *56±2.4* |
| MoteStrain | 79±0.7 | 79±0.7 | 79±0.7 | 53±2.3 | 79±0.7 | *84±1.1* |
| OliveOil | 28±2 | 28±2 | 28±2 | 28±2 | 28±2 | *59±2* |
| OSULeaf | 29±0.7 | 29±1.1 | 29±1.1 | 29±0.9 | 30±0.7 | *45±0.3* |
| Phalanges OutlinesCorrect | 33±3.2 | 33±2.6 | 33±2.6 | 33±2.3 | 33±2.7 | *68±2.4* |
| Plane | 89±2 | 87±1.1 | 87±1.1 | 85±4.3 | 88±1.1 | *98±1.1* |
| ProximalPhalanx OutlineAgeGroup | 18±2 | 18±2.3 | 18±2.3 | 18±1.8 | 18±2.3 | *81±1.9* |
| ProximalPhalanx OutlineCorrect | 36±1.4 | 34±1.1 | 34±1.1 | 33±1.6 | 34±0.9 | *68±1.6* |
| Proximal PhalanxTW | 41±3.9 | 42±4 | 42±4 | 42±4 | 42±3.9 | *53±4.1* |
| Refrigeration Devices | 36±1.8 | 36±1.6 | 36±1.6 | 36±1.3 | 36±1.9 | *43±1.2* |
| ScreenType | 39±1.4 | 38±1.8 | 38±1.8 | 38±1 | 39±1.6 | *36±0.3* |
| ShapeletSim | 50±1.7 | 50±1.4 | 50±1.4 | 49±1.7 | 50±1.4 | *48±0.9* |
| ShapesAll | 49±1.6 | 42±1.1 | 42±1.1 | 43±1.3 | 46±1.8 | *70±0.2* |
| SmallKitchen Appliances | 33±1.4 | 34±1 | 34±1 | 36±1.6 | 34±1.1 | *49±2.2* |
| SonyAIBO RobotSurface | 68±2.6 | 68±2.6 | 68±2.6 | 62±7.3 | 68±2.6 | *68±4.6* |
| SonyAIBO RobotSurfaceII | 81±0.8 | 81±0.8 | 81±0.8 | 71±0.6 | 81±0.8 | *83±0.8* |
| Strawberry | 7±0.3 | 6±0.3 | 6±0.3 | 9±0.7 | 7±0.2 | *96±0.3* |
| SwedishLeaf | 32±1.2 | 26±2.1 | 26±2.1 | 25±0.8 | 29±1.4 | *82±0.3* |
| Symbols | 76±1.5 | 74±1.2 | 74±1.2 | 76±1.4 | 75±1 | *89±0.2* |
| synthetic_control | 80±1.6 | 80±1.7 | 80±1.7 | 37±3.6 | 80±1.6 | *95±1* |
| ToeSegmentation1 | 51±1.5 | 51±1.5 | 51±1.5 | 50±1.2 | 51±1.5 | *57±0.7* |
| ToeSegmentation2 | 63±1.8 | 63±1.8 | 63±1.8 | 55±5.5 | 63±1.8 | *67±3* |
| Trace | 29±2.7 | 29±2.4 | 29±2.4 | 29±2.4 | 29±2.9 | *89±1.8* |
| TwoLeadECG | 45±2.2 | 44±2.3 | 44±2.3 | 37±1.8 | 45±2.2 | *77±0.7* |
| Two_Patterns | 32±1.8 | 31±1.6 | 31±1.6 | 12±0.2 | 31±1.7 | *96±0.4* |
| wafer | 39±1.5 | 39±1.5 | 39±1.5 | 21±1.5 | 39±1.5 | *96±0.9* |
| Wine | 45±0 | 45±0 | 45±0 | 45±0 | 45±0 | *56±0* |
| WordsSynonyms | 40±1.2 | 38±0.5 | 38±0.5 | 32±1 | 39±1.1 | *53±0.4* |
| Worms | 28±0.4 | 27±0.9 | 27±0.9 | 24±1.5 | 28±0.6 | *36±1.2* |
| WormsTwoClass | 49±1.2 | 49±1 | 49±1 | 47±1.4 | 49±1 | *60±1* |

Table 7: Fully Convolutional Network (FCN) classification result on UCR repository under five adversarial attacks.

| FCN | | | | | | |
|---|---|---|---|---|---|---|
| **Datasets** | **FGSM** | **PGD** | **BIM** | **Carlini Wagner** $L^2$ | **MIM** | *No Attack* |
| 50words | 3±0.5 | 6±1.4 | 6±1.4 | 18±3.6 | 4±1.3 | *29±16* |
| Adiac | 5±1.8 | 7±3.8 | 7±3.8 | 11±2.1 | 7±3.5 | *24±17.7* |
| ArrowHead | 40±0 | 14±6.2 | 14±6.2 | 14±6.5 | 15±6 | *80±6.6* |
| Beef | 26±10.2 | 23±9.7 | 23±9.7 | 23±12.7 | 22±7.7 | *52±9.7* |
| BeetleFly | 50±0 | 20±5 | 20±5 | 20±5 | 20±5 | *80±5* |
| BirdChicken | 50±0 | 15±10 | 15±10 | 7±2.9 | 22±2.9 | *94±2.9* |
| Car | 22±0 | 40±27.5 | 40±27.5 | 40±26.2 | 40±25.1 | *47±23.4* |

| | | | | | | |
|---|---|---|---|---|---|---|
| CBF | 83±1.2 | 79±1.6 | 79±1.6 | 1±0.1 | 81±1.3 | *100±0.2* |
| Chlorine Concentration | 39±19.5 | 39±19.8 | 39±19.8 | 38±19.1 | 39±19.8 | *54±18.5* |
| Coffee | 0±0 | 0±0 | 0±0 | 0±0 | 0±0 | *100±0* |
| Computers | 44±10 | 19±5.7 | 19±5.7 | 16±6.1 | 28±11 | *85±6.1* |
| Cricket_X | 16±5.7 | 11±1.8 | 11±1.8 | 13±2.3 | 11±3 | *72±3.7* |
| Cricket_Y | 19±1.9 | 16±3.1 | 16±3.1 | 16±2.9 | 16±3.3 | *69±7.5* |
| Cricket_Z | 13±1.1 | 11±3.2 | 11±3.2 | 14±3.5 | 11±2.1 | *72±5.1* |
| DiatomSize Reduction | 16±4.9 | 6±0.9 | 6±0.9 | 7±0.5 | 7±0.7 | *93±0.7* |
| DistalPhalanx OutlineAgeGroup | 19±4.7 | 19±4.4 | 19±4.4 | 19±4.4 | 19±4.4 | *80±4.3* |
| DistalPhalanx OutlineCorrect | 38±9.6 | 32±6.1 | 32±6.1 | 32±6.2 | 33±6.6 | *69±6.1* |
| Distal PhalanxTW | 15±1.1 | 17±1.2 | 17±1.2 | 17±1.1 | 17±1.1 | *73±2.1* |
| Earthquakes | 36±4.1 | 34±3.2 | 34±3.2 | 25±2.5 | 35±3.3 | *76±2.5* |
| ECG200 | 49±6.5 | 16±3.1 | 16±3.1 | 11±1.8 | 24±5 | *89±1.8* |
| ECG5000 | 69±6.9 | 33±24.7 | 33±24.7 | 4±0.4 | 51±12.5 | *94±0.4* |
| ECGFiveDays | 38±9.5 | 2±0.2 | 2±0.2 | 2±0.3 | 2±0.3 | *99±0.3* |
| ElectricDevices | 43±1.3 | 32±2.7 | 32±2.7 | 14±3.3 | 35±2.9 | *70±3.7* |
| FaceAll | 66±0.7 | 41±0.4 | 41±0.4 | 8±2.7 | 57±0.4 | *90±2.8* |
| FaceFour | 6±2.3 | 3±1.8 | 3±1.8 | 5±1.8 | 3±1.2 | *94±0.7* |
| FacesUCR | 68±2.4 | 40±7.9 | 40±7.9 | 4±0.7 | 56±4.4 | *93±0.8* |
| FISH | 13±0.4 | 19±11.5 | 19±11.5 | 22±11.9 | 18±11 | *60±2.9* |
| Gun_Point | 51±2.7 | 2±0.7 | 2±0.7 | 1±0.4 | 4±2.4 | *100±0.4* |
| Ham | 37±3.4 | 37±3.5 | 37±3.5 | 37±3.5 | 37±3.5 | *64±3.5* |
| Haptics | 23±3.1 | 18±4.8 | 18±4.8 | 19±5 | 18±4.8 | *29±3.4* |
| Herring | 60±0 | 46±8.2 | 46±8.2 | 49±11.9 | 54±5.5 | *60±0* |
| InlineSkate | 16±0.5 | 13±5.2 | 13±5.2 | 16±6.7 | 13±4.5 | *22±7.6* |
| InsectWingbeat Sound | 13±1.8 | 11±1.3 | 11±1.3 | 12±1.5 | 11±1.4 | *23±4.4* |
| ItalyPower Demand | 84±1 | 81±1.7 | 81±1.7 | 5±0.5 | 83±1.5 | *96±0.3* |
| LargeKitchen Appliances | 50±4.9 | 32±23.7 | 32±23.7 | 21±17.5 | 45±13.9 | *74±16* |
| Lighting2 | 40±1.7 | 29±1 | 29±1 | 29±1 | 30±1.7 | *72±1* |
| Lighting7 | 32±7.6 | 19±2.9 | 19±2.9 | 17±3.5 | 23±4.2 | *74±1.6* |
| Meat | 34±0 | 45±13.7 | 45±13.7 | 52±24.9 | 47±11.7 | *34±0* |
| MedicalImages | 23±6.8 | 14±2 | 14±2 | 14±3.1 | 16±1.2 | *77±2.8* |
| MiddlePhalanx OutlineAgeGroup | 18±6.6 | 18±5.9 | 18±5.9 | 17±5.7 | 18±6.1 | *70±6.7* |
| MiddlePhalanx OutlineCorrect | 44±22.5 | 43±21.6 | 43±21.6 | 45±24.2 | 43±21.6 | *58±21.4* |
| MiddlePhalanxTW | 20±10 | 23±11 | 23±11 | 21±9 | 23±10.7 | *48±12.8* |
| MoteStrain | 80±1 | 78±1.2 | 78±1.2 | 10±0.5 | 79±1.5 | *91±0.5* |
| OliveOil | 18±19.3 | 16±21.2 | 16±21.2 | 18±19.3 | 18±19.3 | *56±15.1* |
| OSULeaf | 14±0 | 12±4 | 12±4 | 12±4.4 | 11±4.1 | *75±16.7* |
| Phalanges OutlinesCorrect | 36±2.5 | 36±2.5 | 36±2.5 | 36±2.6 | 36±2.5 | *65±2.6* |
| Plane | 40±5.8 | 11±3.9 | 11±3.9 | 0±0 | 25±6.5 | *100±0* |
| ProximalPhalanx OutlineAgeGroup | 32±23.7 | 22±8.8 | 22±8.8 | 25±10.7 | 22±8.8 | *64±18.9* |
| ProximalPhalanx OutlineCorrect | 32±26.8 | 31±26.4 | 31±26.4 | 31±26.2 | 31±26.8 | *70±26.2* |

| | | | | | | |
|---|---|---|---|---|---|---|
| Proximal PhalanxTW | 18±8.2 | 14±3.1 | 14±3.1 | 15±4.7 | 14±2.9 | *75±2.9* |
| Refrigeration Devices | 40±3.5 | 36±0.9 | 36±0.9 | 35±1.7 | 36±1 | *46±1.7* |
| ScreenType | 33±3.3 | 28±3.6 | 28±3.6 | 27±3.6 | 29±4.3 | *62±5.2* |
| ShapeletSim | 8±3.7 | 8±3.1 | 8±3.1 | 8±2.8 | 8±3.1 | *93±2.8* |
| ShapesAll | 4±1.4 | 3±2.9 | 3±2.9 | 7±0.6 | 3±1.9 | *19±18* |
| SmallKitchen Appliances | 53±16.7 | 37±18.1 | 37±18.1 | 39±22.6 | 41±11.1 | *43±12.3* |
| SonyAIBO RobotSurface | 84±2.2 | 82±2.7 | 82±2.7 | 5±0.3 | 83±2.7 | *97±0.6* |
| SonyAIBO RobotSurfaceII | 86±1.5 | 84±2.1 | 84±2.1 | 3±0.5 | 85±1.7 | *98±0.5* |
| Strawberry | 44±20.8 | 31±8.8 | 31±8.8 | 31±8.9 | 31±9.1 | *70±8.8* |
| SwedishLeaf | 28±1.7 | 10±2.6 | 10±2.6 | 6±3.6 | 13±3.3 | *93±3.6* |
| Symbols | 36±3.2 | 6±1.6 | 6±1.6 | 5±0.6 | 15±1.9 | *94±1.3* |
| synthetic_control | 95±1 | 95±1.3 | 95±1.3 | 3±0.9 | 95±1.2 | *98±0.7* |
| ToeSegmentation1 | 41±6.2 | 11±0.8 | 11±0.8 | 3±0.7 | 18±3 | *98±0.7* |
| ToeSegmentation2 | 43±1.4 | 26±2.3 | 26±2.3 | 14±2.8 | 36±0.5 | *87±2.8* |
| Trace | 52±18.6 | 18±8.9 | 18±8.9 | 1±0.6 | 43±2.9 | *100±0.6* |
| TwoLeadECG | 7±3.1 | 2±0.4 | 2±0.4 | 1±0.1 | 3±0.7 | *100±0.1* |
| Two_Patterns | 34±7.3 | 15±0.7 | 15±0.7 | 15±0.7 | 19±2.3 | *86±0.7* |
| wafer | 8±3.2 | 3±0.9 | 3±0.9 | 1±0.2 | 3±1.3 | *100±0.2* |
| Wine | 50±0 | 50±0 | 50±0 | 50±0 | 50±0 | *50±0* |
| WordsSynonyms | 5±2.2 | 9±3.3 | 9±3.3 | 12±1.5 | 6±1.9 | *30±10.2* |
| Worms | 17±1.7 | 21±3.6 | 21±3.6 | 21±5.3 | 21±3.4 | *48±7.3* |
| WormsTwoClass | 48±5 | 39±2.3 | 39±2.3 | 39±2.5 | 40±4.2 | *62±2.3* |

Table 8: ResNet classification result on UCR repository under five adversarial attacks.

| ResNet | | | | | | |
|---|---|---|---|---|---|---|
| **Datasets** | **FGSM** | **PGD** | **BIM** | **Carlini Wagner** $L^2$ | **MIM** | *No Attack* |
| 50words | 8±2.3 | 10±1 | 10±1 | 13±1.5 | 9±1.5 | *67±0.7* |
| Adiac | 5±0.2 | 10±1.2 | 10±1.2 | 10±0.2 | 10±0.4 | *82±0.7* |
| ArrowHead | 34±11.5 | 13±0.9 | 13±0.9 | 13±1.5 | 15±1 | *79±2.3* |
| Beef | 24±8.9 | 19±5.1 | 19±5.1 | 18±3.9 | 22±3.9 | *74±3.4* |
| BeetleFly | 29±5.8 | 17±5.8 | 17±5.8 | 17±5.8 | 17±5.8 | *84±5.8* |
| BirdChicken | 54±5.8 | 14±2.9 | 14±2.9 | 14±2.9 | 20±5 | *87±2.9* |
| Car | 20±1 | 9±4.5 | 9±4.5 | 8±3.9 | 10±4.9 | *89±3.5* |
| CBF | 89±1.4 | 87±1.8 | 87±1.8 | 1±0.2 | 88±1.6 | *100±0.2* |
| Chlorine Concentration | 14±0.4 | 14±0.8 | 14±0.8 | 13±0.4 | 14±0.7 | *82±1.1* |
| Coffee | 0±0 | 0±0 | 0±0 | 0±0 | 0±0 | *100±0* |
| Computers | 58±5.4 | 24±1.3 | 24±1.3 | 20±3.2 | 45±5.1 | *82±2.6* |
| Cricket_X | 33±3 | 17±2.5 | 17±2.5 | 14±2.1 | 27±1.9 | *76±2.4* |
| Cricket_Y | 23±0.6 | 13±0.7 | 13±0.7 | 13±0.6 | 16±1.7 | *80±1.1* |
| Cricket_Z | 28±2.9 | 14±2 | 14±2 | 13±0.8 | 22±2.4 | *78±1.4* |
| DiatomSize Reduction | 10±4.1 | 4±1.5 | 4±1.5 | 5±2 | 4±1.4 | *97±1.9* |
| DistalPhalanx OutlineAgeGroup | 18±2.4 | 17±1.8 | 17±1.8 | 17±2 | 17±1.8 | *81±1.8* |
| DistalPhalanx OutlineCorrect | 29±3.6 | 23±1 | 23±1 | 21±1.2 | 25±1.7 | *80±1* |

| | | | | | |
|---|---|---|---|---|---|
| Distal PhalanxTW | 15±0.3 | 15±0.8 | 15±0.8 | 14±0.6 | 15±0.9 | *76±0.7* |
| Earthquakes | 48±2.9 | 45±2.7 | 45±2.7 | 24±1 | 46±3.1 | *80±1.2* |
| ECG200 | 69±4.4 | 50±11.6 | 50±11.6 | 13±2.1 | 63±4.1 | *88±2.4* |
| ECG5000 | 73±0.8 | 61±1.3 | 61±1.3 | 5±0.3 | 66±1.3 | *94±0.3* |
| ECGFiveDays | 33±16.2 | 4±1.6 | 4±1.6 | 3±0.6 | 6±3.8 | *98±0.7* |
| ElectricDevices | 41±2.1 | 31±1.7 | 31±1.7 | 15±2.4 | 36±2.2 | *70±4.5* |
| FaceAll | 76±0.4 | 69±1 | 69±1 | 11±0.5 | 74±0.7 | *83±1.6* |
| FaceFour | 30±5.2 | 9±2.4 | 9±2.4 | 4±2.9 | 22±3.5 | *95±0.7* |
| FacesUCR | 74±1.4 | 64±2.3 | 64±2.3 | 3±0.8 | 70±1.6 | *95±0.4* |
| FISH | 13±0 | 3±0.9 | 3±0.9 | 3±1.2 | 3±0.9 | *98±1* |
| Gun_Point | 23±5.6 | 6±2 | 6±2 | 1±0.4 | 10±0.7 | *100±0* |
| Ham | 30±2.9 | 29±2 | 29±2 | 30±2.4 | 29±2 | *72±2* |
| Haptics | 20±0.2 | 22±3.1 | 22±3.1 | 21±3.7 | 21±3.8 | *49±4* |
| Herring | 49±11 | 41±1 | 41±1 | 41±1 | 41±1 | *60±1* |
| InlineSkate | 15±1.3 | 19±2 | 19±2 | 19±2.9 | 19±1.9 | *32±3.1* |
| InsectWingbeat Sound | 22±0.5 | 23±0.6 | 23±0.6 | 23±0.4 | 24±0.3 | *46±1.1* |
| ItalyPower Demand | 87±1.3 | 86±0.8 | 86±0.8 | 7±0.9 | 86±1.3 | *97±0.2* |
| LargeKitchen Appliances | 59±2.8 | 32±2.7 | 32±2.7 | 8±1.4 | 47±1.2 | *90±0.8* |
| Lighting2 | 46±0 | 42±2.6 | 42±2.6 | 27±1.7 | 43±1.7 | *74±1.7* |
| Lighting7 | 36±3.7 | 20±4.2 | 20±4.2 | 19±2.1 | 24±7.7 | *74±4.2* |
| Meat | 17±15.5 | 8±5.4 | 8±5.4 | 8±5.4 | 8±5.4 | *93±5.4* |
| MedicalImages | 47±5 | 28±3.8 | 28±3.8 | 15±2.5 | 36±2.4 | *78±0.7* |
| MiddlePhalanx OutlineAgeGroup | 16±1.5 | 16±0.7 | 16±0.7 | 15±0.2 | 16±0.8 | *75±1* |
| MiddlePhalanx OutlineCorrect | 27±9.1 | 27±9 | 27±9 | 27±9.1 | 27±9 | *74±9.2* |
| Middle PhalanxTW | 15±2.8 | 17±0.4 | 17±0.4 | 17±0.6 | 17±0.7 | *62±0.8* |
| MoteStrain | 76±0.9 | 73±1.1 | 73±1.1 | 10±0.8 | 75±1.1 | *91±0.8* |
| OliveOil | 14±0 | 17±5.8 | 17±5.8 | 18±3.9 | 17±5.8 | *79±2* |
| OSULeaf | 14±1 | 6±2.2 | 6±2.2 | 5±1.9 | 6±2.2 | *94±2.8* |
| Phalanges OutlinesCorrect | 27±3 | 17±0.9 | 17±0.9 | 18±0.7 | 17±0.9 | *84±0.9* |
| Plane | 73±6.2 | 41±6.4 | 41±6.4 | 0±0 | 63±5.3 | *100±0* |
| ProximalPhalanx OutlineAgeGroup | 16±4.8 | 15±0.8 | 15±0.8 | 16±1.5 | 15±0.8 | *86±0.6* |
| ProximalPhalanx OutlineCorrect | 16±2.6 | 11±1.6 | 11±1.6 | 11±1.7 | 11±1.6 | *90±1.6* |
| Proximal PhalanxTW | 8±1.2 | 13±0.5 | 13±0.5 | 14±0.3 | 14±0.4 | *82±0.5* |
| Refrigeration Devices | 35±2.5 | 34±3.1 | 34±3.1 | 31±2.3 | 34±3.1 | *54±0.6* |
| ScreenType | 35±7 | 29±2.6 | 29±2.6 | 28±3.5 | 32±4.5 | *61±3.8* |
| ShapeletSim | 13±7.9 | 12±8.6 | 12±8.6 | 10±10.2 | 13±8.1 | *91±9.9* |
| ShapesAll | 7±0.7 | 3±0.3 | 3±0.3 | 5±0.3 | 4±0.7 | *88±0.5* |
| SmallKitchen Appliances | 44±4.5 | 28±5.6 | 28±5.6 | 29±7.7 | 34±5.2 | *56±16* |
| SonyAIBO RobotSurface | 80±2.3 | 79±2.9 | 79±2.9 | 14±3.2 | 79±2.5 | *92±0.9* |
| SonyAIBO RobotSurfaceII | 81±1.1 | 79±1.6 | 79±1.6 | 4±0.8 | 80±1 | *98±0.8* |
| Strawberry | 24±16 | 22±17.7 | 22±17.7 | 22±17.6 | 22±17.7 | *80±17.6* |

| | | | | | | |
|---|---|---|---|---|---|---|
| SwedishLeaf | 34±0.8 | 16±0.5 | 16±0.5 | 4±0.5 | 22±0.9 | *96±0.4* |
| Symbols | 32±2.1 | 8±0.5 | 8±0.5 | 5±1.6 | 16±1.5 | *95±1.7* |
| synthetic_control | 95±0.7 | 95±0.4 | 95±0.4 | 20±4 | 95±0.7 | *100±0.4* |
| ToeSegmentation1 | 54±1.8 | 31±2.5 | 31±2.5 | 4±0.7 | 39±2 | *97±0.7* |
| ToeSegmentation2 | 45±5.2 | 35±5.9 | 35±5.9 | 11±2.5 | 41±4.3 | *90±2.5* |
| Trace | 30±2.1 | 13±9.7 | 13±9.7 | 2±1.6 | 37±8.6 | *98±0* |
| TwoLeadECG | 8±4.8 | 2±0.6 | 2±0.6 | 1±0.5 | 4±1.7 | *100±0.3* |
| Two_Patterns | 68±1.9 | 42±6.2 | 42±6.2 | 6±1.1 | 56±3.8 | *96±1* |
| wafer | 17±11.8 | 7±7.8 | 7±7.8 | 2±0.2 | 11±10.8 | *100±0.1* |
| Wine | 34±16 | 25±8.4 | 25±8.4 | 25±8.4 | 25±8.4 | *76±8.4* |
| WordsSynonyms | 15±3.1 | 14±1 | 14±1 | 16±0.4 | 14±1.4 | *54±1.3* |
| Worms | 26±2 | 21±1.5 | 21±1.5 | 19±0.9 | 25±0.4 | *63±2* |
| WormsTwoClass | 54±2.7 | 29±2 | 29±2 | 27±2 | 32±1.4 | *75±1.4* |

