# OpenReview forum: "Evaluating the Robustness of Time Series Anomaly and Intrusion Detection Methods against Adversarial Attacks"
_ICLR.cc/2022/Conference — ICLR 2022 Submitted_

### Official Review · Reviewer_iy4H · 2021-10-24

**Correctness:** 3
**Technical Novelty And Significance:** 2
**Empirical Novelty And Significance:** 2
**Recommendation:** 5
**Confidence:** 5

**Main Review:**

(this review comes from a security-focused reviewer) I liked the research direction of the paper which I believe is worthy of investigation. Specifically, the claim that the paper is the first to investigate adversarial attacks against time-series-based ML systems in the context of “anomaly and intrusion detection” is valid. I also appreciated that the attacks were carried out on multiple state-of-the-art systems trained on distinct datasets.

However, I am not impressed by the achieved results and, moreover, I believe that the paper falls short to be of adequate “impact” for ICLR. Let me elaborate on the abovementioned weaknesses.

First, the **poor treatment of previous work**. It is true that adversarial attacks against time-series-based ML applications are not well-investigated. Yet, within the context of Intrusion Detection, there exist a plethora of works that highlighted the vulnerability of ML-based systems to adversarial attacks. A very recent summary is [A], which reports more than 40 papers within the specific context of Network Intrusion Detection (similar reviews can also be found in [B] and [C], although they also cover Malware and Phishing detectors). Specifically, [D] in 2018 was one of the first papers to expose the weaknesses of ML-based NIDS to imperceptible adversarial perturbations. Hence, the authors’ claim that “While adversarial attacks have been extensively studied in the context of image recognition, they have not been extensively investigated for anomaly and intrusion detection systems.” (start of page 2) is wrong, and should be amended.

**Unimpressive Results**. I acknowledge that the paper is “the first”; however we are now in 2021, and the fact that ML-systems can be broken via adversarial perturbations is an accepted fact. Although in the specific context of time-series analyses such papers are rare, this is simply another setting. In other words: from a purely ML-perspective, there is no difference between a “static” and “temporal” phenomenon: in a “static” problem the dimension of the feature set is smaller as each sample is denoted only by its “current” values; in a “temporal” problem the feature set is larger and more complex as it also takes into account the past history. However, aside from such “dimensionality difference”, the two problems are identical. As such, I am not surprised that a white box attacker can thwart a similar system with a FGSM and PGD attack. To aggravate the issue, the attacker has no bounds on the perturbations: I acknowledge that this is a “realistic” assumption, however the attacker essentially can do whatever they want. In these circumstances, after thousands of paper that showed that ML is susceptible to adversarial attacks, is there any surprise that a “yet another” ML system can be broken?

**Poor Threat Model**. This issue is strongly related to the previous comment, but since the targeted system is specifically devoted to security tasks it is even more relevant. In other words: the attacker knows and can do everything with the target system. I could accept a similar assumption if the target system addressed a task that had little in common with security (e.g., the general computer vision problem). However, when the targeted model is meant to be part of a security system, the situation changes. Specifically, an attacker that has full control of a security system is a violation of the basic security assumptions, as any security system under attacker’s control is compromised and its security is not guaranteed. In a similar circumstance, the attacker could achieve the same (or even worse) results without relying on adversarial examples.

**Inadequate problem definition**. According to the authors, the attacks targeted systems devoted to “anomaly and intrusion detection methods”. However, little is done to explain these problems. Specifically, “anomaly detection” is a very broad term. It is true that the authors attack state-of-the-art systems, yet I would have appreciated a little more context w.r.t. what are the “anomalies” that are meant to be detected. Moreover, the authors’ definition of “intrusion detection” is extremely limited, as the attack seems to be specific to CAN (controller area network) which are systems embedded in vehicles, and thus represent a very niche subfield of intrusion detection. The resulting impression is that the paper simply assumes a (unrealistic and “obvious”) threat model, and then carries out (obviously successful) attacks but without considering any real world constraints or implications. I believe that a similar research method is inadequate in 2021, and even more so for ICLR. For instance, a common issue with current adversarial strategies is that the adversarial samples are crafted without taking into account the “problem space” ([E]). After reading the paper I was not able to figure out if this is also the case for the considered attacks.

EXTERNAL REFERENCES

[A]: "Modeling Realistic Adversarial Attacks against Network Intrusion Detection Systems." ACM Digital Threats: Research and Practice (2021).

[B]: "Adversarial machine learning applied to intrusion and malware scenarios: a systematic review." IEEE Access 8 (2020): 35403-35419.

[C]: "Addressing adversarial attacks against security systems based on machine learning." 2019 11th International Conference on Cyber Conflict (CyCon). Vol. 900. IEEE, 2019.

[D]: "Evading botnet detectors based on flows and random forest with adversarial samples." 2018 IEEE 17th International Symposium on Network Computing and Applications (NCA). IEEE, 2018.

[E]: "Intriguing properties of adversarial ml attacks in the problem space." 2020 IEEE Symposium on Security and Privacy (SP). IEEE, 2020.



**Summary Of The Paper:**

The paper tackles the problem of adversarial attacks against time-series-based ML applications devoted to intrusion detection. The paper is relatively simple: they use existing adversarial ML strategies (white-box attacks) to thwart a similar ML system. The main contribution is the fact that few efforts investigated adversarial attacks against time-series based ML methods, and – specifically – no paper considered “anomaly and intrusion detection” scenarios.

Overall, the presentation of the paper is adequate.
The quality of the English text is fair.
Figures and Tables are appropriate.
The topic addressed by the manuscript is relevant and within ICLR’s scope.
The references are not appropriate.
The contribution is not very significant.

STRENGTHS:
+ It is truly the first
+ Evaluation on multiple datasets


WEAKNESSES
- Poor treatment of previous work
- Unimpressive results
- Poor threat model
- Inadequate problem definition


**Summary Of The Review:**

The paper has its merit, but the poor and unrealistic threat model undermine the “impact” of the results. It is true that attacks against time-series based ML are rare, but many efforts studied the impact of “traditional” adversarial attacks in intrusion detection. The extremely powerful adversary is able to thwart state-of-the-art ML systems… but such outcome is obvious considered the attacker’s assumptions.

In the current state, I do not believe the paper passes the bar for ICLR. I believe the paper requires a more realistic threat model that clearly outlines the real capabilities and knowledge of the adversary. In its current state, despite some originality, the paper represents just a “yet another adversarial ML paper”. The complete lack of references to security-focused works is emblematic of the issues affecting this paper. I acknowledge that ICLR is not a security-focused venue, but since the claimed contribution of this paper is its "novel" application, then I expect a stronger security background.

---

> ### Author Response · Authors · 2021-11-13
> **Thank you for providing an in-depth analysis of our work from a security point of view.**
>
> We appreciate the reviewer's in-depth analysis of our work from a security standpoint. By fixing the reviewer's concerns has undoubtedly strengthened our paper. In the following paragraphs, we attempt to address the majority of these concerns and/or our rationales for doing so.
>
> >**Poor treatment of previous work**
>
> We appreciate the reviewer pointing out this inconsistency. We concur with the reviewer that there are numerous papers on adversarial attacks in the intrusion detection space. We will undoubtedly incorporate them into our related work and amend the statement to reflect this. However, we believe that adversarial attacks are still under-investigated in the anomaly detection domain, and, more importantly, their significance is under-publicized in the anomaly detection domain. Nonetheless, we will temper our assertion and make the necessary changes.
>
> >**Unimpressive Results**
>
> We fully acknowledge the reviewer's concerns. Additionally, we concur with the reviewer's assessment of static and temporal phenomena. Our experiments produced predictable results, and it's unsurprising that the attack broke the anomaly detection systems. However, we believe that it is critical to demonstrate this, particularly in the domain of anomaly detection, where the majority of SOTA detectors do not consider adversarial attacks to be a threat and thus do not discuss any vulnerabilities of their proposed systems against them.
>
> We believe that our paper is not particularly significant in the broader machine learning context, as there are numerous adversarial attacks on machine learning systems. It is critical, however, in the context of anomaly detection, and our work aims to lay the groundwork in this direction.
>
> >**Poor Threat Model**
>
> We thank the reviewer for bringing this to our attention. This is an excellent point that should be pursued further. We have been oblivious to it from this vantage point. We concur that if the targeted model is a component of a security system, assuming that the attacker can do anything is a dangerous assumption. We made this assumption in our work because the architecture, training method, and other characteristics of the evaluated methods are publicly available, and because, as is the case with the Korean Aerospace Research Institute (KARI), a model very similar to the one published by Tariq et al. [i] was deployed. As a result, generating a single attack against the public version has a good chance of succeeding against the private version (e.g., the one deployed at KARI).
>
> Nonetheless, it is not apparent how the attacker injects adversarial examples into the system. We excluded it from the scope of our paper because each method discussed in our paper is unique, and it would require extensive research to determine the best way for the attacker to gain access to the system or to inject adversarial examples into the actual data stream. We will clarify and extend the threat model to include this information in the final version.
>
> >**Inadequate problem definition**
>
> We appreciate you bringing this critical point to our attention. We will most certainly include additional context for anomalies. The anomaly detection datasets used in our work (SMAP, MSL, KARI, and SMD) contain a variety of different types of anomalies that fall into three major categories: point anomalies, contextual anomalies, and collective anomalies. SOTA detectors are typically developed to detect all three types of anomalies.
>
> Our work was primarily focused on anomaly detection, but we included a side experiment on intrusion detection methods (CAN) to demonstrate that similar results are achievable in both domains. This will be clarified in the main text, and it will be made abundantly clear that our primary focus is on anomaly detection systems. We have not considered the work of Pierazzi et al. [E]. We appreciate the reviewer's input and pointing us in the right direction. We will consider how to incorporate problem space [E] into our methodology in the future.
>
> We concur that our work omitted some important real-world constraints. Due to two reasons: 1) It is difficult to determine how these systems are deployed; for example, for the KARI dataset, we later discovered that their CLMPPCA method is deployed on a separate server from the rest of the workflow, which makes it less secure than other components of the pipeline, such as the control system. 2) Because each model is unique and deployed differently, writing a separate constraint for each evaluated method would have made it more difficult for the reader to follow. However, we concur with the reviewer that this may have oversimplified the threat model.
>
> **Reference**
>
> [i]. Tariq, Shahroz, et al. "Detecting anomalies in space using multivariate convolutional LSTM with mixtures of probabilistic PCA." Proceedings of the 25th ACM SIGKDD International Conference on Knowledge Discovery & Data Mining. 2019.

---

### Official Review · Reviewer_VZJZ · 2021-10-31

**Correctness:** 3
**Technical Novelty And Significance:** 2
**Empirical Novelty And Significance:** 2
**Recommendation:** 5
**Confidence:** 4

**Main Review:**

Strengths:
- The problem is relevant
- Clearly outlined criteria for choosing anomaly detection baselines
- I strongly appreciate Section 9, in addition to the authors making all their code available
- Authors avoid base-rate fallacy by using F1, Precision, Recall

Weaknesses:
- There is no new strong methodology/attack proposed
- The authors claim to be the first tackling adversarial in intrusion detection, but it is not the case (e.g., [1-3] )
- The authors only consider feature-space attack, where it could have been more interesting to consider realizable problem-space perturbations [4] which consider modifications of real world objects/traffic within CANs

Comments:

The paper proposes a thorough evaluation of some time series-based anomaly detectors applied on datasets in the CAN domain. While the evaluation is generally well thought, I do have some concerns with the work:
- **Contribution**. The contribution seem rather limited: at the end of the day, you are just applying a PGD/FGSM datasets to attack time series-based anomaly detection systems that use neural networks. It is not necessarily surprising that they are weak against this scenario. Moreover, you claim to be the first tackling the problem of adversarial attacks within the context of intrusion detection. This may unwittingly be a problem of overclaiming: there has been several works in the past (e.g., [1-3]) which studied the problem of adversarial attacks in the context of network intrusion detection systems, which are a super-set of your considered CAN-based attacks.
- **Challenges**. There seems to be no new attack proposed or no significant challenge in applying PGD/FGSM attacks to these type of detectors. Hence, related to the prior point, it is unclear how much significant are the findings of this work.
- **Problem-space**. You seem to be working only in the feature space, whereas it would have been more interesting to explore how modifying real-world CAN traffic properties would impact the effectiveness or complexity of the adversarial attack. In particular, you could have considered a modeling based on the framework proposed by Pierazzi et al [4] about realizable problem-space attacks.
- **Focus on CAN**. It is partially unclear why the paper narrows the focus on CAN-only datasets. It seems to be a somewhat convenient choice, whereas there would be other DDoS datasets in more general network settings. Unless there is a strong motivation to focus only on CAN, it is partially unclear why the authors restricted themselves to this domain. I am saying this because the proposed attack does not depend on the CAN domain. This may change if you consider problem-space attacks. It is also important that the focus of CAN, if preserved, is clarified since the very beginning. As the abstract/intro of the paper seem to be tackling more general IDS systems.

References:
- [1] Apruzzese, Giovanni, and Michele Colajanni. "Evading botnet detectors based on flows and random forest with adversarial samples." 2018 IEEE 17th International Symposium on Network Computing and Applications (NCA). IEEE, 2018.
- [2] Apruzzese, Giovanni, Michele Colajanni, and Mirco Marchetti. "Evaluating the effectiveness of adversarial attacks against botnet detectors." 2019 IEEE 18th International Symposium on Network Computing and Applications (NCA). IEEE, 2019.
- [3] Corona, Igino, Giorgio Giacinto, and Fabio Roli. "Adversarial attacks against intrusion detection systems: Taxonomy, solutions and open issues." Information Sciences 239 (2013): 201-225.
- [4] Pierazzi, Fabio, et al. "Intriguing properties of adversarial ml attacks in the problem space." 2020 IEEE Symposium on Security and Privacy (SP). IEEE, 2020.


EDIT:
Sligthly raising my score after rebuttal discussion with authors.

**Summary Of The Paper:**

This paper shows adversarial attacks to time series anomaly and IDS systems, specifically focused on CAN (Controlled Area Network) dataset. The authors devise some criteria to select datasets and algorithms, and apply PGD/FGSM based attacks and show the brittleness of the detection systems.

**Summary Of The Review:**

I feel the major issue is with the limited contributions and originality of the work, and the unclear focus on CAN-only attacks, when the proposed methodology does not seem to tackle this domain specifically.

---

> ### Author Response · Authors · 2021-11-10
> **We value your thoughtful and perceptive comments. We provide clarification on CAN dataset and address each of your concerns in detail.**
>
> To begin, we would like to express our gratitude to **R2** for reviewing our manuscript and providing us with valuable feedback.
>
> **Clarification:** Before we address the reviewer's concerns in detail, we'd like to make one clarification. Our research is not limited to the CAN domain. Our work focuses on two problems: **anomaly detection** and **intrusion detection**; to accomplish this, we analyzed data from three distinct time series domains:
> 1) Resource utilization of 38 compute cluster machines (**SMD** dataset).
> 2) Three aerospace industry datasets (**MSL**, **SMAP**, **KARI**).
> 3) Controller Area Network for vehicles (**OTIDS** dataset).
> ---
> ---
> ---
>
> >_**Contribution.** The contribution seem rather limited: at the end of the day, you are just applying a PGD/FGSM datasets to attack time series-based anomaly detection systems that use neural networks. It is not necessarily surprising that they are weak against this scenario. ..... , which are a super-set of your considered CAN-based attacks._
>
> We concur with the reviewer in that we are using PGD and FGSM attacks on Neural Networks (NNs), and the results are predictable. However, these findings should be highlighted, particularly in the domain of anomaly detection, which is the primary focus of our paper, because the majority of recently published SOTA anomaly detection methods do not include or discuss their method's vulnerability to adversarial attacks.
>
> Additionally, our contribution does not end with the application of PGD and FGSM attacks to NNs and demonstrating their vulnerability. We provide a framework for future authors to evaluate their methods against all adversarial attacks included in the latest version of the _CleverHans_ library. As an illustration of this streamlined framework, we present in the **Appendix** several adversarial attacks on various NNs (MLP, FCN, ResNet) trained on 71 datasets from the *UCR repository*, including **_FGSM, PGD, BIM, Carlini Wagner L2, and MIM_**. In the future, the authors can simply "_plug and play_" their detectors to evaluate them against a variety of adversarial attacks using this framework.
>
> Moreover, we would like to express our gratitude to the reviewer for pointing out several pertinent prior works on adversarial attacks in the intrusion detection domain. We will incorporate them into our final version and also temper our assertions of being the first.
>
> >_**Challenges.** There seems to be no new attack proposed or no significant challenge in applying PGD/FGSM attacks to these types of detectors. Hence, related to the prior point, it is unclear how much significant are the findings of this work._
>
> We wholeheartedly agree that, in the absence of a new attack, the findings in this paper appear to be insignificant. However, we want to emphasize that if simple attacks such as FGSM and PGD are quite effective against anomaly and intrusion detection methods. Then, more sophisticated attacks will almost certainly outperform simpler ones. To illustrate our central point that "_future anomaly and intrusion detection methods should consider robustness against adversarial examples_," we do not require a sophisticated and complex attack, as simple attacks can already significantly reduce performance.
>
> >_**Problem-space.** You seem to be working only in the feature space, whereas it would have been more interesting to explore ..... the framework proposed by Pierazzi et al [4] about realizable problem-space attacks._
>
> We would like to thank the reviewer for suggesting an intriguing future direction for this research. We will most certainly incorporate it into the final version of this paper as a future direction.
>
> >_**Focus on CAN.** It is partially unclear why the paper narrows the focus on CAN-only datasets. ..... As the abstract/intro of the paper seem to be tackling more general IDS systems._
>
> It is unclear whether the reviewer is referring to the intrusion detection method itself or to the entire paper. The method for detecting intrusions is focused on the CAN dataset. We outline our justifications below. However, it would be incorrect to assert that our paper focuses exclusively on the CAN dataset or IDS, as we incorporated data from a variety of domains, as follows:
>
> 1. For anomaly detection, we use datasets from the aerospace industry (i.e., **SMAP**, **MSL**, and **KARI**) and datacenter resource utilization to detect anomalies (**SMD**).
> 2. For intrusion detection, we used the **CAN** dataset.
>     - _Reason_: The CAN dataset was chosen due to its utility to the automobile industry. We concur with the reviewer that the same attacks can be used against models trained on other intrusion detection datasets, including **DARPA**, **KDD**, **NSL-KDD**, and **ADFA-LD**. Additionally, the authors have a greater familiarity with the Controller Area Network domain, which influenced their choice of the CAN dataset, as it aided in our ability to comprehend and analyze the results of the CAN experiments.

---

### Official Review · Reviewer_hY71 · 2021-11-01

**Correctness:** 4
**Technical Novelty And Significance:** 2
**Empirical Novelty And Significance:** 2
**Recommendation:** 5
**Confidence:** 4

**Main Review:**

Pros:

The paper clearly shows that DNN and GNN based anomaly and intrusion detection methods are sensitive to adversarial attacks. The authors consider all sota methods they can find and demonstrate their findings with thorough experiments. The paper is well written.


Cons:
1.	This paper is not the first paper to transfer adversarial attacks to anomaly detection models for time series. As noticed by the authors, the paper titled “Robustness of Autoencoders for Anomaly Detection Under Adversarial Impact” published at IJCAI 2020 has studied the impact of IFGSM attacks on autoencoder-based anomaly detection methods. Therefore, it is better not to highlight “first” in the abstract.
2.	The major contribution, that is, the performance of DNN and GNN based anomaly and intrusion detection methods suffers from the adversarial attacks, is not surprising. The potentially more attractive perspective of this research, as mentioned by the authors in Section 6, is to propose more robust algorithms, which is not done here.

Minor Comments:
1.	Page 9, CANTransfer Performance: please provide some insights on why CANTransfer is more robust to FGSM attacks.
2.	Page 5, Section 4, Datasets, the last sentence: Table 1 summarize -> summarizes
3.	Page 5, Section 4, Evaluation Metrics: the prediction, recall, and f1-score -> the precision, recall, and f1-score
4.	The same sentence as above: from thresholding methods -> from the thresholding methods
5.	Page 6, Line 3: I synthetic -> (i) synthetic


**Summary Of The Paper:**

In this paper, the authors apply adversarial attack techniques, such as FGSM and PGD, to “fool” the SOTA DNN and GNN methods for anomaly and intrusion detection in time series. They show that small perturbations to the input time series can lead to significant deterioration in the performance of the SOTA methods.

**Summary Of The Review:**

It is a contribution to show that current SOTA anomaly and intrusion methods are quite sensitive to adversarial attacks through a systematic set of experiments. However, no new algorithms or methods or insights are proposed in this paper, and so I lean to rank the paper as borderline.

---

> ### Author Response · Authors · 2021-11-10
> **We appreciate your thoughtful and insightful comments. We will address your concerns point by point.**
>
> >_This paper is not the first paper to transfer adversarial attacks to anomaly detection models for time series. As noticed by the authors, the paper titled “Robustness of Autoencoders for Anomaly Detection Under Adversarial Impact” published at IJCAI 2020 has studied the impact of IFGSM attacks on autoencoder-based anomaly detection methods. Therefore, it is better not to highlight “first” in the abstract._
>
> We appreciate the reviewer's suggestion. The "_Robustness of Autoencoders for Anomaly Detection Under Adversarial Impact_" paper considered only autoencoders, whereas we evaluated networks of any style. However, we concur with the reviewer that it is preferable not to highlight it in the abstract. We will temper our assertion.
>
> ---
>
> >_The major contribution, that is, the performance of DNN and GNN based anomaly and intrusion detection methods suffer from the adversarial attacks, is not surprising. The potentially more attractive perspective of this research, as mentioned by the authors in Section 6, is to propose more robust algorithms, which is not done here._
>
> We acknowledge the reviewer’s comment. Developing a defense against these adversarial time series attacks is an intriguing area of research. We did not pursue it further in this paper, however, for the following reasons:
>
> - ✨ We wanted to concentrate on and highlight the weaknesses of the current anomaly and intrusion detection system in this paper. Therefore, detection methods in the future should treat adversarial attacks as legitimate threats and demonstrate resilience to them. Additionally, the findings of our work could be cited in the future during the peer review process to question the authors about the robustness of their new detection method against such adversarial attacks. On a lighter note, most anomaly detection papers submitted to ICLR 2022 continue to ignore adversarial attacks. Our goal with this work is to emphasize their importance in developing robust systems.
>
> - ✨ As discussed briefly in the paper, adversarial training can be used to mitigate such attacks. However, as **Tramer et al.** [A] point out in "_On Adaptive Attacks to Adversarial Example Defenses_," this is not a foolproof strategy, and **no single defense is adequate against all types of adaptive attacks**. As a result, we believed it would be difficult to incorporate both offensive and defensive strategies into a single paper. As a result, we excluded it from the scope of this paper and will write another article focusing exclusively on the defense portion, as it is a lengthy subject in and of itself.
>
> **Reference**
>
> [A].  Florian Tramer, Nicholas Carlini, Wieland Brendel, and Aleksander Madry. On adaptive attacks to adversarial example defenses. In H. Larochelle, M. Ranzato, R. Hadsell, M. F. Balcan, and H. Lin (eds.), Advances in Neural Information Processing Systems, volume 33, pp. 1633–1645. Curran Associates, Inc., 2020. URL https://proceedings.neurips.cc/paper/2020/file/11f38f8ecd71867b42433548d1078e38-Paper.pdf.
>
>
> ---
>
> >**Minor Comments**
> >1. Page 9, CANTransfer Performance: please provide some insights on why CANTransfer is more robust to FGSM attacks.
> >2. Page 5, Section 4, Datasets, the last sentence: Table 1 summarize -> summarizes
> >3. Page 5, Section 4, Evaluation Metrics: the prediction, recall, and f1-score -> the precision, recall, and f1-score
> >4. The same sentence as above: from thresholding methods -> from the thresholding methods
> >5. Page 6, Line 3: I synthetic -> (i) synthetic
>
> We appreciate the reviewer's highlighting the grammatical errors. We will make the proper corrections in the final version of this paper.

---

> ### Comment · Reviewer_hY71 · 2021-11-26
> **Ack**
>
> Appreciate the authors provide detailed response.
> However, there are some unresolved points such as how about attack transferability, what is the impact if some general adversarial defense methods (adversarial training, robust filtering like de-noise auto-encoder).
> The insight given by this paper is limited and not comprehensive under its current status.

---

> > ### Author Response · Authors · 2021-11-29
> > **Thank you**
> >
> > We would like to express our gratitude to the reviewer for their insightful comments and suggestions, such as attack transferability and basic defense methods. We intend to include them in a future version of this work.

---

### Decision · Program_Chairs · 2022-01-20

**Decision:**

Reject

**Comment:**

The paper investigates attacks against time series analysis methods such as GNN and DNN for anomaly and intrusion detection. Standard attacks such as FGSM and PGD are extended for the time series domain and evaluated on several datasets including automotive, aerospace and resource utilization datasets. While the authors claim to be the first to investigate such attacks, some related work was not considered in the paper, which was pointed out by reviewers. Also some other weaknesses of the proposed method, e.g., its focus on feature space perturbations were pointed out. Hence, while acknowledging the importance and the novelty of this paper's contributions, the reviewers agree that the paper must be better positioned in the context of the related work in order to be accepted.

---

> ### Public Comment · ~Khaza_Anuarul_Hoque1 · 2022-06-15
> **This work is NOT novel and there exists already published works (2020) in adversarial attacks for (multivariate) time series (regression) with codes on GitHub**
>
> We found this submission by accident and, but we are indeed surprised to see that the authors claimed that this as the “first work” in adversarial robustness analysis (they disguised the works as anomaly detection though) for time series. However, this claim is NOT true, and it is not possible that the authors have done this by a mistake (as google easily find the previous works when you put the keyword "time series and adversarial example"). We have already published our works in multivariate time series robustness analysis for regression 2 years ago in IEEE ICMLA 2020 and IEEE AIPR 2020 and the authors very cleverly avoided mentioning those works from our group. The authors have also mentioned in the limitations section that “It is hard to reproduce the same results as demonstrated by the paper, mainly when the codes are not from the original authors but developed by the community.” This is also not true, our FGSM and BIM attacks for multivariate time series regression is available on github for last 2 years (look at our paper, we have provided the URL unlike these authors). It is very disturbing to find such unethical academic practice to avoid citing published works to give an impression that the submission under review is 'novel’  to the reviewers.
>
> Our papers from 2020 on time series adversarial attacks for regression (with codes):
> 1. Mode, Gautam Raj, and Khaza Anuarul Hoque. "Adversarial examples in deep learning for multivariate time series regression." 2020 IEEE Applied Imagery Pattern Recognition Workshop (AIPR). IEEE, 2020.
>
> 2. Mode, Gautam Raj, and Khaza Anuarul Hoque. "Crafting adversarial examples for deep learning based prognostics." 2020 19th IEEE International Conference on Machine Learning and Applications (ICMLA). IEEE, 2020.

---

> > ### Public Comment · ~Shahroz_Tariq1 · 2022-06-20
> > **Please be respectful and read the paper carefully. It is not directly related to regression but more towards classification in anomaly and intrusion detection.**
> >
> > First, it would be great if you showed more respect toward your academic fellows. In addition, this article has previously been rejected, and a rewritten version is being reviewed at another venue. Thus, there is no need for additional feedback on this outdated work. However, I will attempt to spend some time clarifying.
> >
> >  - They disguised the works as anomaly detection though) for time series.
> >    - We did not disguise it. This is a paper about anomaly and intrusion detectors on time series, which is distinct from regression problems (look at the definition of each and also **classification vs. regression**). If you read the reviews below, you'll notice that the reviewers have identified some missing prior work on **"adversarial attacks on intrusion detection"** that is more directly related to our work than yours, and we have acknowledged that we may have missed some relevant work. Also, if your work was so **essential** to ours, one of the reviewers or the meta-reviewers should have rejected our work on that basis or at least mentioned it, but they did not, which tells us something. I am not suggesting that the reviewers are all-knowing, and neither are we; thus, afford us the benefit of the doubt. Moreover, since there has been prior work on the **adversarial attack on intrusion detection** (as noted by reviewers), we have omitted the intrusion detection portion of this work in the future version (which is undergoing review at another venue). Now, we are just concerned with the anomaly detection part.
> >
> > - It is not possible that the authors have done this by a mistake (as google easily find the previous works when you put the keyword "time series and adversarial example") ..... and the authors very cleverly avoided mentioning those works from our group.
> >   - Words such as cleverly avoiding and disguising may reveal your mindset, but please do not have the same opinion of us. Even though your papers are not directly connected to our problem as it is a regression-related adversarial attack paper, and we concentrated on the adversarial attack on anomaly detection, which is more of a **classification problem**, still we have cited numerous "**adversarial attack papers on time-series**" studies that are directly relevant to our work. Please read the manuscript attentively to see, for example, that we have mentioned Fawaz et al. Nonetheless, let's assume that your articles are directly relevant to ours; have you considered that the authors may have overlooked your paper while searching for related work? Please refrain from being too pessimistic and beginning to believe that we avoided, etc. Overall, because we are not addressing regression tasks, your study may not have been examined; otherwise, there was no harm in it for us in citing your work. Furthermore, we do not know you or your group, so why would we attempt to avoid your group's work? In addition, we believe in **equal opportunity**; thus, the group, nationality, or country does not matter to us; if the work is relevant, we will cite it.
> >
> > Again, I am requesting more respect from you. I can understand that you believed your work to be relevant, even though we continue to believe it is not directly connected; it must have irritated or discouraged you that we did not cite it, but there are other effective ways to say this. These type of statements does not accurately represent our academic culture. Lastly, we will look at your papers again and cite them if they seem relevant to our current draft.
> >
> > **We will not make any further comments as this is an outdated and rejected draft.**
> >
> > Thank you!